# *Drosophila* insulator proteins exhibit in vivo liquid–liquid phase separation properties

Bright Amankwaa, Todd Schoborg , Mariano Labrador

**Mounting evidence implicates liquid–liquid phase separation (LLPS), the condensation of biomolecules into liquid-like droplets in the formation and dissolution of membraneless intracellular organelles (MLOs). Cells use MLOs or condensates for various biological processes, including emergency signaling and spatiotemporal control over steady-state biochemical reactions and heterochromatin formation. Insulator proteins are architectural elements involved in establishing independent domains of transcriptional activity within eukaryotic genomes. In *Drosophila*, insulator proteins form nuclear foci known as insulator bodies in response to osmotic stress. However, the mechanism through which insulator proteins assemble into bodies is yet to be investigated. Here, we identify signatures of LLPS by insulator bodies, including high disorder tendency in insulator proteins, scaffold–client–dependent assembly, extensive fusion behavior, sphericity, and sensitivity to 1,6-hexanediol. We also show that the cohesin subunit Rad21 is a component of insulator bodies, adding to the known insulator protein constituents and γH2Av. Our data suggest a concerted role of cohesin and insulator proteins in insulator body formation and under physiological conditions. We propose a mechanism whereby these architectural proteins modulate 3D genome organization through LLPS.**

## Introduction

It is becoming increasingly clear that the establishment of independent higher order DNA domains (3D genome organization) in eukaryotes plays a role in important aspects of genome function, including replication, transcription, and DNA damage repair (Lupiáñez et al, 2015; Ulianov et al, 2016; Stam et al, 2019; Sanders et al, 2020). The 3D genome organization comprises the distinct nuclear spaces occupied by chromosomes known as chromosome territories, which are in turn made up of active and inactive DNA folds referred to as A and B compartments, respectively. At high resolution, the genome is organized in contiguous regions characterized by high interaction frequencies called topologically associating domains (TADs), which are separated by boundaries that limit the interactions between these domains. TAD domains are well conserved and are proposed to delimit regulatory landscapes where functional interactions between gene promoters and distal regulatory elements occur (Rao et al, 2014; Lupiáñez et al, 2015; Szabo et al, 2020; Torosin et al, 2020).

Suggested mechanisms in the generation of these genomic features include transcription, phase separation, and loop extrusion (Banigan et al, 2020; Kentepozidou et al, 2020). Even though the contributions of these processes appear to differ across species, the involvement of certain architectural proteins is crucial and evolutionarily conserved. Insulator-binding proteins (IBPs), lamins, transcription factors, and the cohesin complex notably belong to these architectural proteins (Matthews & White, 2019; Rowley et al, 2019). Canonically, IBPs are assembled on DNA elements known as insulators to shield gene promoters from promiscuous interactions with enhancers in a process referred to as enhancer blocking (Kyrchanova et al, 2013). In addition, they serve as physical barriers that prevent heterochromatin spreading to active regions (Özdemir & Gambetta, 2019). Most of the insulator proteins, including Suppressor of Hairy wing (Su(Hw)), centrosomal protein 190 (Cp190), modifier of mdg4 67.2 (Mod(mdg4)67.2), and the *Drosophila* CCTC-binding factor (dCTCF), have been identified in *Drosophila* (Raab et al, 2012; Özdemir & Gambetta, 2019). In contrast, CTCF is the only IBP characterized in mammals so far (Raab et al, 2012; Özdemir & Gambetta, 2019). On the other hand, cohesins are proteins found in all eukaryotes and are traditionally known to mediate sister chromatid cohesion and homologous recombination during cell division, in addition to their role in transcription (Nasmyth & Haering, 2009). The cohesin complex forms a ring structure consisting of a structural maintenance of chromosome protein dimer (SMC1/SMC3) bridged by the Rad21 protein. They are loaded onto chromosomes by the Nipped-B (Scc2, Mis4, NIPBL)–Mau2 (Scc4) complex and removed by the Pds5-Wapl (Rad61) complex and separase before anaphase during the cell cycle (Dorsett, 2019).

Insulator and cohesin proteins synergistically mediate the formation of TADs through a chromatin looping process known as loop extrusion in mammals (Banigan et al, 2020; Kentepozidou et al,

---

Department of Biochemistry and Cellular and Molecular Biology, The University of Tennessee, Knoxville, TN, USA

Correspondence: labrador@utk.edu
Todd Schoborg's present address is Department of Molecular Biology, University of Wyoming, Laramie, WY, USA.

2020). The loop extrusion model posits that the ring-shaped cohesin complex extrudes loops by threading chromatin and therefore bringing distant DNA sites into spatial proximity, thereby favoring certain enhancer–promoter interactions (Banigan et al, 2020; Kentepozidou et al, 2020). According to this model, the insulator protein CTCF serves as a barrier for the extrusion through a convergent orientation-dependent DNA binding. Consistent with this, the deletion of individual CTCF sites in the DNA allows long-range contacts between genomic regions normally belonging to separated TADs with sometimes pathological implications, including abnormal limb development and cancer (Soshnikova et al, 2010; Lupiáñez et al, 2015; Nora et al, 2017; Ibrahim & Mundlos, 2020). Even though IBPs and cohesin overlap substantially in *Drosophila*, to our knowledge, it has not been accepted that DNA loop extrusion plays a major role in *Drosophila* spatial genome organization (Nuebler et al, 2018). In addition, the *Drosophila* homolog of CTCF (dCTCF) does not pair to form loop domains and is not preferentially found at TAD boundaries (Rowley et al, 2017; Wang et al, 2018b). It is also worth noting that, even in mammals, not all TADs can be explained by the loop extrusion model (Rao et al, 2014; Hansen, 2020).

It has been suggested that liquid–liquid phase separation (LLPS) drives the *Drosophila* genome organization and complements the loop extrusion process in mammals, especially with respect to TAD formation (Ulianov et al, 2016; Rowley et al, 2017; Feric & Misteli, 2021). LLPS is a fundamental physicochemical process of de-mixing biomolecules to form a distinct concentrated phase that lies in equilibrium with a less concentrated phase (Shin & Brangwynne, 2017; Murthy & Fawzi, 2020). LLPS mediates the formation of a myriad of biological condensates including the nucleolus, stress granules, paraspeckles, and p-bodies (Brangwynne et al, 2009, 2011; Oliver et al, 2010; Wheeler et al, 2016). In addition, several lines of evidence indicate that phase separation modulates the segregation of the eukaryotic genome into active and inactive compartments (Rudolph et al, 2007; Strom et al, 2017; Larson & Narlikar, 2018; Falk et al, 2019; Stam et al, 2019; Shakya et al, 2020). This is supported by the liquid-like droplet formation by the genome-associated proteins, heterochromatin protein 1α (HP1α) (Lawrimore & Bloom, 2019) and the cohesin subunit SMC in yeast (Ryu et al, 2021). The regulatory hub formation of super-enhancers, transcription factors, the mediator complex, and RNA polymerase are also proposed to be LLPS driven (Wutz et al, 2017; Boeynaems et al, 2018; Nuebler et al, 2018; Sabari et al, 2018; Stam et al, 2019).

Remarkably, proposals that the *Drosophila* genome organization is predominantly mediated by LLPS do not address the question of the role that insulator proteins may play in such organization. It was initially held in the field that multiple IBPs bound to insulator sites coalesce to form hubs that served as contact sites for organizing the *Drosophila* 3D genome (Gerasimova et al, 2000; Labrador & Corces, 2002; Byrd & Corces, 2003). It was proposed that such hubs appeared under the microscope as the foci identified as insulator bodies (Pai et al, 2004; Capelson & Corces, 2005). However, the existing literature at the time did not address the specific biological mechanisms that would mediate the coalescence of chromatin and IBPs into insulator body structures. Our laboratory first addressed this issue by demonstrating that insulator bodies, defined as the large foci observed under the microscope, only form during the

osmotic stress response and during apoptosis (Schoborg et al, 2013; Schoborg & Labrador, 2014). We showed that increasing salt concentration to 250 mM in the media leads to the amalgamation of all insulator proteins into insulator bodies. This process is concomitant with the sumoylation of Cp190, a significant reduction of IBPs binding to chromatin (measured fluorescence microscopy and by ChIP) and to a significant decrease in long-range genome interactions as measured by chromosome conformation capture (3C). Though results from these experiments suggested that insulator proteins contribute to long-range interactions in the genome, we showed that the large foci known as insulator bodies are only induced as a response to osmotic stress and are not significantly attached to chromatin (Schoborg et al, 2013; Schoborg & Labrador, 2014). More recently, results from our lab show that the phosphorylated histone variant H2Av (γH2Av) interacts with IBPs at insulator sites genome-wide and that γH2Av is also a critical component of insulator bodies (manuscript submitted for publication).

Here, we consider the hypothesis that *Drosophila* insulator bodies are formed through phase separation by analyzing their condensate behaviors and by extension we ask whether insulator proteins also functionally associate forming condensates when bound to chromatin under normal physiological conditions. To the best of our knowledge, insulator bodies have not been assessed for hallmark features that support LLPS so that it remains unknown whether IBPs form insulator bodies via LLPS under physiological conditions. In this work, by analyzing the sequence determinants of various *Drosophila* insulator proteins and the sensitivity of the bodies to 1,6-hexanediol, we propose that the clustering of IBPs into bodies is mediated through both electrostatic, hydrophobic and/or π–contact interactions. In addition, we provide evidence that insulator proteins exhibit a significant degree of LLPS properties, both as insulator bodies under salt stress and at physiological conditions. In light of our results, we speculate that *Drosophila* insulator proteins mediate their functions through LLPS.

## Results

### *Drosophila* IBPs display a high disorder tendency and show weak polyampholyte properties

Multiple folded domains, posttranslational modifications, and intrinsic disorderness contribute to the multivalency of proteins needed for LLPS (Alberti et al, 2019; Owen & Shewmaker, 2019; Perdikari et al, 2021). Among these traits, intrinsic disorderness appears to be the strongest predictor of a protein's phase separating abilities and has been the most consistent feature in constituents of biomolecular condensates (Mészáros et al, 2018; Alberti et al, 2019). Indeed, mutations in disordered domains are frequently observed in diseases associated with LLPS dysregulation (Vacic et al, 2012; Darling et al, 2019). Intrinsically disordered regions (IDRs) encompass low-complexity regions (LCRs), that is, protein domains in which particular amino acids are overrepresented compared with the amino acid proportions found in the proteome (Necci et al, 2018). Using two IDR prediction tools, IUPred2 (Mészáros et al, 2018) and Predictors of Natural Disordered Regions (Peng et al, 2006),

we demonstrate that the *gypsy* chromatin insulator core complex proteins Su(Hw), Mod(mdg4)67.2, and Cp190 have high disorder propensity (Fig 1A). For example, about 67.5%, 47.1%, and 57.3% lengths of Cp190, Su(Hw), and Mod(mdg4)67.2, respectively, are predicted to be disordered (Fig S1A). The conserved dCTCF insulator protein also showed similar disorder tendency with about 52% of its length being disordered (Fig S1A). Interestingly, the combined disorder scores of known insulator body constituents and other IBPs are comparable to the scores of experimentally verified cases of LLPS *Drosophila* proteins curated in *PhaSepDB* (Figs 1C and S1A and B). *PhaSepDB* is a novel database that provides a collection of manually curated phase separation–related proteins (You et al, 2020). This implies that the structural disorder found in insulator proteins is no different from those of known phase separation proteins in *Drosophila*.

Different flavors of IDRs exist based on specific protein features deemed as the driving forces of LLPS by promoting weak multivalent interactions (Krishnakumar & Kraus, 2010; Pak et al, 2016; Alberti et al, 2019). These features are used in phase separation algorithms to predict a specific protein's propensity to form condensates (You et al, 2020; Shen et al, 2021). For instance, LARKS (low-complexity aromatic-rich kinked segments) uses 3D profiling to measure the probability of a given sequence to bind weakly to each other by forming a pair of kinked $\beta$-sheets (Hughes et al, 2018), PScore relies on the $\pi$–$\pi$ contact tendency of residues in a given protein sequence (Vernon et al, 2018), whereas R + Y depends on the number of tyrosine and arginine residues within disordered regions of proteins (Wang et al, 2018a). We compared the PScore of CP190, Su(Hw), and Mod(mdg4)67.2 to those of the well-characterized phase separation proteins FUS, TDP43, and hnRNPA2 using the PSP website (Chu et al, 2022). The IBP PScores were comparable to those of FUS, TDP43, and hnRNPA2 (Fig S2A). Based on the reliance of the Pscore algorithm on $\pi$–$\pi$ contact interactions, these predictions would mean the tested IBPs have high proportions of aromatic ring amino acids (e.g., histidine, tyrosine, phenylalanine, and tryptophan) (Vernon et al, 2018; Vernon & Forman-Kay, 2019). In addition, residues with $\pi$ bonds on their side chains (e.g., glutamic acid, aspartic acid, asparagine, arginine, and glutamine) and small residues with exposed backbone peptide bonds (e.g., proline, threonine, glycine, and serine) can also exhibit $\pi$–$\pi$ interactions (Vernon et al, 2018). We however ruled out the possibility of aromatic residues as a relatively lower number of LARKS were recorded for the gypsy-associated IBPs using the database LARKSdb (Hughes et al, 2021) (Fig S2B). This denotes that these IBPs do not rely on kink-forming amino acids like glycine and the aromatic residues. We inferred that the nonaromatic residues glutamic acid, aspartic acid, asparagine, arginine, and glutamine may play crucial roles in the IDR and hence, LLPS properties of IBPs.

Recent reports demonstrate a correlation between the density of charged residue tracts and IDR conformations that can distinguish distinct condensates (Das et al, 2015; Holehouse et al, 2017). Analysis of their amino acid distribution showed that IBPs generally depict multiple uncompensated charged residues (Figs 1B and S3). Specifically, at least one-fifth of the amino acids in the sequence of the core gypsy insulator proteins, as well as in dCTCF and BEAF32 are charged residues, including aspartate, glutamate, arginine, and lysine (Fig S3A). These translate into an overall net charge per

residue (NCPR) of −0.09, −0.024, −0.01, −0.03, and −0.01 for CP190, Mod(mdg4)67.2, Su(Hw), dCTCF, and BEAF32, respectively, implying a less mixed amino acid charge distribution (Fig S3A). NCPR expresses the difference between the fractions of positively (f+) and negatively (f−) charged residues (Holehouse et al, 2017). Proteins with a preponderance of charged residues such as those found in IBPs are demonstrated to undergo phase separation through electrostatic interactions (Pak et al, 2016). The strong likelihood of electrostatic-mediated clustering of insulator proteins can be explained by the suggestion that unlike stretches of residues in which charges are uniformly dispersed, tracts of contiguous charged residues provide weak electrostatic forces that contribute to phase separation (Somjee et al, 2020).

As for IBPs the f+ ≈ f− and the NCPR are close to zero, IBPs generally typify as "polyampholytes" (Pappu et al, 2008). Indeed, a representation on the Das-Pappu's phase diagram of IDP/IDR ensembles show that the insulator body constituents ($\gamma$H2Av, Su(Hw), CP190, Mod(mdg4)67.2), and dCTCF lie between weak polyampholytes or weak polyelectrolytes (R1) and strong polyampholytes (R3) that form non-globular conformations (Fig 1D). A number of studies show that this almost electrical neutrality enables polyampholytes to collapse, whereas uneven charges lead to structural expansion because of repulsive forces (Srivastava & Muthukumar, 1996; Das et al, 2015; Holehouse et al, 2017). We therefore infer that electrostatic interactions between the segments of conformationally heterogeneous IBPs provide a differential attraction, leading to their assembly into condensates.

### Insulator bodies are liquid droplets and not solid aggregates

Despite the apparent contribution of electrostatic interactions, it has been shown elsewhere that at high salt concentrations, electrostatic interactions are screened out leaving hydrophobic interactions to drive phase transition (Krainer et al, 2021). Therefore, to obtain further insights into the nature of the chemical interactions underlying assembly of insulator proteins, we looked at the effect of 1,6-hexanediol (1,6-HD) on insulator bodies. 1,6-HD is an agent that perturbs hydrophobicity-dependent LLPS condensates presumably through disruption of weak hydrophobic interactions (Kroschwald et al, 2017; Sabari et al, 2018; Lesne et al, 2019). In addition, unlike LLPS entities like the nucleolus (Vertii et al, 2019) and transcription condensates (Boehning et al, 2018), solid aggregates such as viral replication compartments (McSwiggen et al, 2019a), the cytoskeleton (Kroschwald et al, 2017), and tetO binding (Ryu et al, 2021) are largely resistant to 1,6-HD. To determine whether insulator bodies are sensitive to 1,6-HD, we exposed insulator bodies to 1,6-hexanediol. After induction of osmotic stress, cells were incubated with 5% 1,6-HD in 250 mM NaCl for 2 min, fixed, and immuno-stained with anti-Su(Hw) and anti-Cp190. The number of insulator bodies and the colocalization of Su(Hw) with CP190 were determined in a quantitative manner by fluorescence microscopy and imaging analysis (see the Materials and Methods section). The minimal time of exposure and the low 1,6-hexanediol concentration were to prevent any deleterious effect of hexanediol on the cells, including hyper-condensation of chromatin as reported elsewhere (Itoh et al, 2021). Results show that insulator bodies are highly sensitive to 1,6-HD, illustrated by the drastic

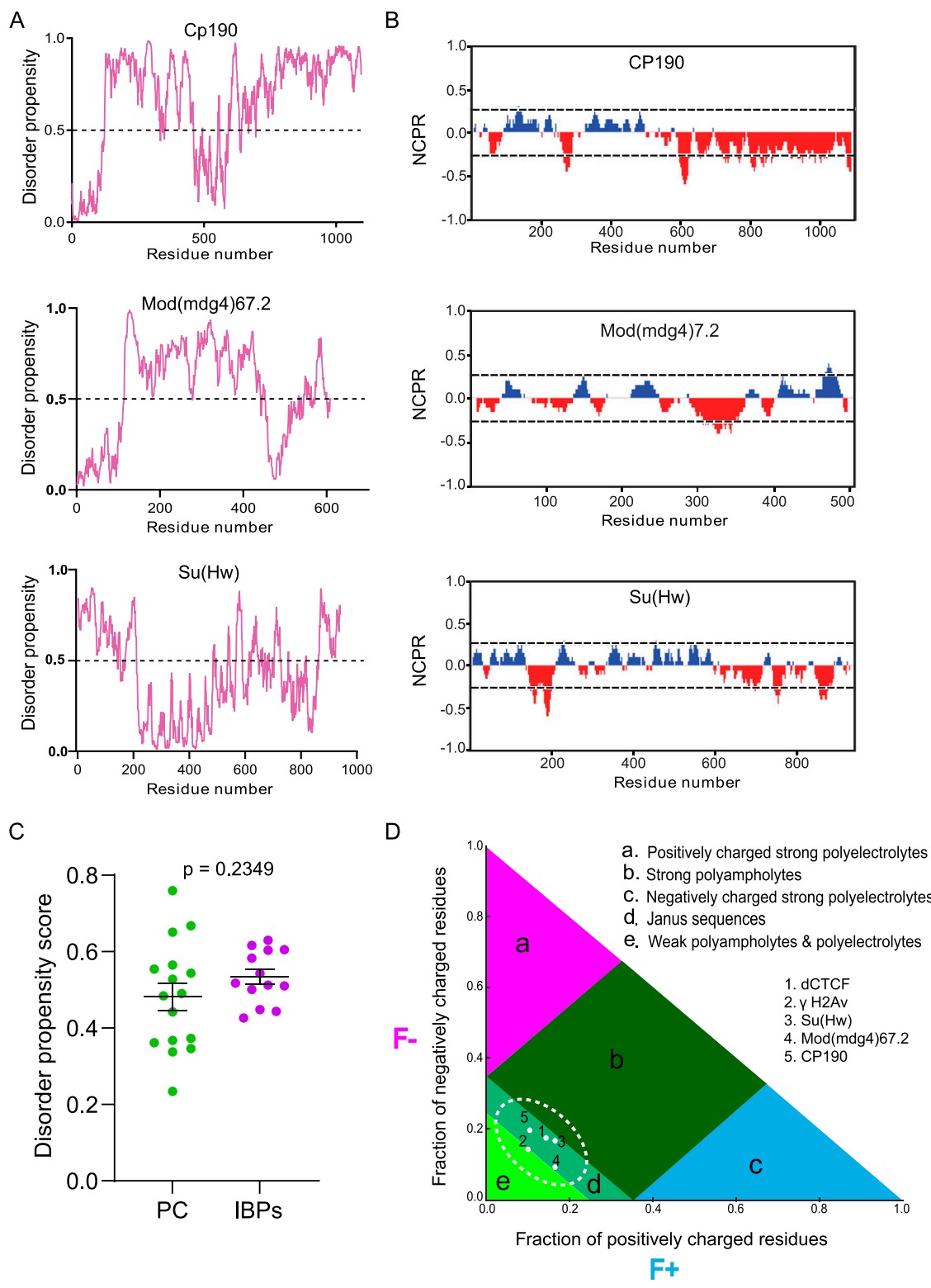

**Figure 1.** ***Drosophila* insulator-binding proteins (IBPs) display a high disorder tendency and show weak polyampholyte properties.**
**(A)** Analysis of the intrinsic disorderness of insulator proteins. A score higher than 0.5 (indicated with broken lines) denotes a high probability of disorder. Top, Cp190; middle, Mod(mdg4)67.2; bottom, Su(Hw). **(B)** Partitioning of insulator proteins into 20 overlapping segments or blobs. Positively charged residues (blue peaks); negatively charged residues (red peaks); nonpolar residues (gaps). The x-axis denotes net charge per residue. The y-axis denotes residue positions. **(C)** Comparison of disorder propensity scores of PhaSepDB-curated *Drosophila* proteins denoted as "PC." Total number of PCs, n = 16 and IBPs denoted as "IBPs." Total number of IBPs, n = 13. **(D)** Das-Pappu's phase diagram showing likely insulator protein disordered conformations. F(−), fraction of negatively charged residues; F (+), fraction of positively

reduction in the number of foci per cell (Fig 2A and B) and the pronounced reduction in the colocalization between Su(Hw) and Cp190 in the bodies (Fig 2A and C).

## Insulator bodies undergo fusions to form enlarged circular structures

Formation of spherical structures and fusion behaviors are striking features of LLPS-mediated condensates (Kroschwald et al, 2015; Alberti et al, 2019). The sphericity of these condensates is explained by surface tension-driven reductions at the boundary between the dilute and condensed phases (Hyman et al, 2014). To test whether insulator bodies are spherical structures akin to those LLPS-driven condensates, time-lapse microscopy of stress-induced insulator bodies in *Drosophila* S2 cells was analyzed using GFP-tagged Su(Hw) in *Drosophila* S2 cells. We used a circularity value of 1.0 to indicate a perfect circle and an approach toward 0.0 as an increasingly elongated polygon as used elsewhere (Takashimizu & Iiyoshi, 2016) to quantify the spherical nature of the Su(Hw)-associated insulator bodies. As expected for liquid-like droplet state, insulator bodies showed a characteristic circular shape with median circularity of 0.89 (Fig 3A and B). As a form of control, an mCherry-tagged BEAF-32 protein previously demonstrated to form an oval shape halo around the insulator bodies was significantly less circular (Fig S4).

Insulator bodies showed marked closeness and fusion events resulting in enlarged condensate formation (Fig 3A). This is consistent with both the size (Fig 3C) and number (Fig 3D) of insulator bodies increasing with time and roughly plateauing later (about 30 s), presumably after a threshold concentration is reached upon salt exposure. Obviously visible in the data are fusion events via coalescence. Ostwald ripening, the dissolution of small liquid bodies in favor of the growth of larger liquid bodies, is not apparent as a prominent mode of droplet coarsening (Voorhees, 1992).

## Insulator bodies exhibit scaffold–client properties

Though LLPS condensates typically harbor a plethora of proteins, their structural integrity hinges on a small subset of proteins referred to as scaffolds (Alberti et al, 2019). Other components are rather passively recruited into the condensates and hence are called "clients" (Song et al, 2020). Client proteins are dispensable but become enriched through interactions and affinity with the scaffold (Decker et al, 2007; Banani et al, 2017; Ditlev et al, 2018; Zhang et al, 2019). We therefore sought to find out which among the three core gypsy insulator proteins could be serving as scaffolds or clients in insulator bodies. We generated insulator bodies by salt-stressing wing imaginal disc cells from third instar *Drosophila* larvae (as explained above) in mutant backgrounds of Cp190, Su(Hw), and Mod(mdg4)67.2. We then quantified the number of Cp190-, Mod(mdg4)67.2-, and Su(Hw)-associated insulator bodies in

the mutant backgrounds of each of these proteins. Interestingly, in either Cp190 or Mod(mdg4)67.2 mutants, we found a significant reduction of Su(Hw)-associated insulator bodies (Figs 4A and B and S5A and B). However, the absence of Su(Hw) does not seem to influence the number of bodies formed by either Cp190 or Mod(mdg4)67.2 (S5C and S5D). Also, both Cp190 and Mod(mdg4)67.2 have similar insulator body reducing effect on each other, implying that they are mutually essential for insulator body formation.

According to the stickers-and-spacers model, phase separation of biomolecules is influenced by specific adhesive individual residue types or short motifs ("stickers") within scaffold proteins (Martin et al, 2020). The model posits that stickers contribute to the main interaction potential and are interspersed by "spacer" elements that influence the ability of the biomolecule to interact with the solvent. Judging from the overall NCPR and the polyampholyte properties displayed by insulator body constituents (Fig 1), we reasoned that the negative amino acid–rich regions could serve as stickers in insulator body scaffolds. To test this, we used combinations of three Cp190 mutants, Cp190$^{P11}$, Cp190H$^{31-2}$, and Cp190$^{4-1}$ which are null, removal of all non-BTB domains, and removal of the glutamic acid–rich region, respectively (Fig 4C). Although wild-type Cp190 has a net charge of residue (NCPR) of –0.09, trans heterozygote of Cp190$^{4-1}$/Cp190$^{P11}$ and Cp190$^{H31-2}$/Cp190$^{P11}$ have NCPRs of +0.03 and –0.02, respectively. Wing imaginal discs from flies expressing mutant Cp190 devoid of the non-BTB domains (Cp190$^{H31-2}$/Cp190$^{11}$) led to a significant reduction in number of insulator bodies (Fig 4D and E). Cells expressing mutant Cp190 devoid of just the glutamic-rich region however showed similar insulator body number to that of the wild type (Fig 4D and E). Though, these results cannot decouple insulator body effect of the truncated Cp190 domain from just the reduction in the NCPR, these results reemphasize the possibility of the negatively charged residues function as stickers in the Cp190 scaffold. An unambiguous attribution of the reduction of the insulator body number to lowered NCPR would warrant targeted shuffling of the charged residues.

## Insulator proteins possess LLPS features at physiological conditions

Next, we asked whether the intrinsic LLPS properties we described in insulator proteins allow IBPs to form condensates in association with chromatin under normal conditions. Analysis of the distribution of insulator proteins before and after osmotic stress has previously revealed the presence of small speckles under physiological conditions (Schoborg et al, 2013). These speckles are significantly smaller and more abundant than the insulator bodies resulting from osmotic stress response. The presence of these speckles in the absence of osmotic stress suggests the possibility that insulator proteins can form condensates either as constitutively formed under normal conditions or as insulator bodies in response to salt stress. Similar observations have been

charged residues. Protein sequences in regions "a" and "c" depict strong polyelectrolyte features with FCR > 0.35 and net charge per residue > 0.3. Such proteins mostly exhibit coil-like conformations. Region "b" corresponds to strong polyampholytes that form distinctly non-globular conformations, such as coil-like, hairpin-like, or hybrids. Region "e" relates to either weak polyampholytes or weak polyelectrolytes that form globule or tadpole-like conformations. Region "d" denotes a continuum of all the possibilities of conformations adopted by proteins in regions "b" and "e." *P*-values < 0.05 are deemed significant.

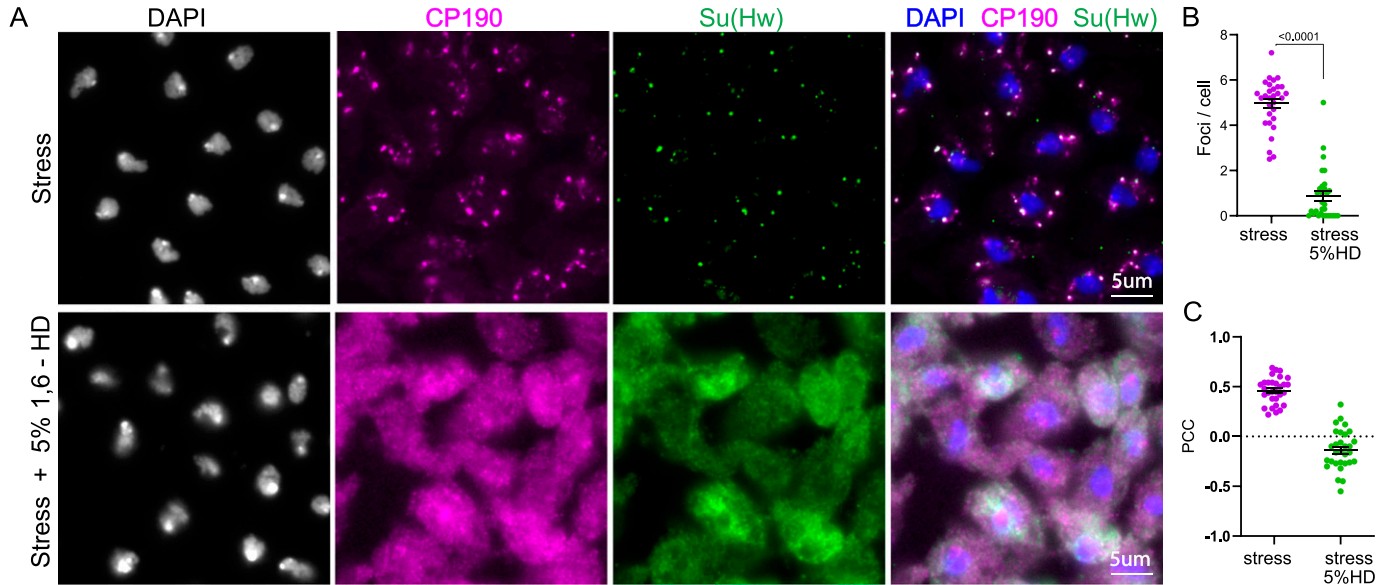

**Figure 2. Insulator bodies are liquid droplets and not solid aggregates.**
**(A)** Insulator bodies formed in response to osmotic stress (Top panel) are dissolved upon treatment with 1, 6-hexanediol (bottom panel). **(B)** The number of insulator bodies are significantly reduced in the presence of 1, 6-hexanediol. **(C)** The level of Pearson correlation (PCC) between CP190 and Su(Hw). Insulator protein signal is plotted with each point representing insulator bodies of wing imaginal discs of one image. In all, 30 images were taken for each treatment. For each treatment, three correlated biological replicates were combined. Statistical differences were determined using unpaired two-tailed *t* test, and *P*-values < 0.05 are deemed significant.

independently reported by others (Buxa et al, 2016). This notion, which implies that without stress insulator proteins may possess intrinsic LLPS properties allowing them to functionally coalesce into condensates, may have important genome organization implications.

To examine this, we first looked at the response of the core gypsy insulator protein bands on polytene chromosomes to 1,6-hexanediol. Polytene chromosomes result from several consecutive rounds of genome replication without cell division in the salivary gland cells of third instar *Drosophila* larva (Zhimulëv, 1996). This polytenization process results in giant chromosomes containing up to 2,000 genome copies per cell, making their structure and morphology easy to analyze under the light microscope. Notably, polytene chromosomes faithfully reproduce the 3D structure and function found in their diploid chromosome counterparts (Eagen et al, 2015; Schwartz & Cavalli, 2017). We reasoned that if insulator proteins phase-separate at their genome-binding sites, the classic insulator protein bands observed in polytene chromosomes would in fact correspond to amplified nucleic-acid/protein condensates. We treated polytene chromosomes from third instar larvae with 5% 1,6-hexanediol followed by immunostaining with Cp190 and Su(Hw) proteins. Results show that both CP190 and Su(Hw) intensities are reduced after incubation with 1,6-hexanediol (Fig 5A and B). Line scans spanning the entire polytene chromosomes show a loss in band sharpness and colocalization of CP190 and Su(Hw) upon exposure to 1,6-hexanediol (Fig 5E and F).

We inferred from these results that the cognate binding of insulator proteins to chromatin is mediated in part by LLPS. If this is true, insulator protein bands on polytene chromosomes should have a level of dynamicity as the rapid turnover of condensate constituents is a key criterion to define LLPS bodies (Alberti et al,

2019; Yoshizawa et al, 2020). Hence, we looked at the dynamic nature of Su(Hw) protein on polytene chromosomes. Su(Hw) is a well-established multi-zinc finger DNA-binding protein (Soshnev et al, 2012), and so, we aimed at determining whether diffusion contributes to its interaction with the DNA not just the binding or reaction kinetics. It is expected that the dependence of fluorescence recovery on the sizes of the bleached area after photobleaching implies both diffusion and binding (diffusion-coupled), whereas the opposite (diffusion-uncoupled) is true for interactions mediated through only binding (Sprague & McNally, 2005; McSwiggen et al, 2019b). To determine this, we expressed Su(Hw)::EGFP in *Drosophila* polytene chromosomes with vestigial GAL4 driver (Simmonds et al, 1995; Schoborg et al, 2013). Using laser confocal microscopy, the recovery of both large (1.60 × 1.0 *μm*) and small (1.0 × 0.6 *μm*) oval Su(Hw)::EGFP spots on the polytene chromosomes were analyzed. Our assessment is that the recovery of Su(Hw) polytene bands after photobleaching depends on the spatial scale as different spot sizes displayed different recovery patterns (Fig 5G and H). At an arbitrary time of 25 s after bleaching, the percent recoveries for the large and small spots are 56% and 36%, respectively. The implication is that diffusion and binding are intertwined throughout the measured recovery phase. This implies that the Su(Hw) complex association with the DNA is partly diffusion-mediated and not just from the strong structured domain interactions.

### The cohesin subunit RAD21 colocalizes with *Drosophila* insulator proteins and is an insulator body constituent

Individual biological condensates can encapsulate hundreds of distinct molecular components. For example, the nucleolus for

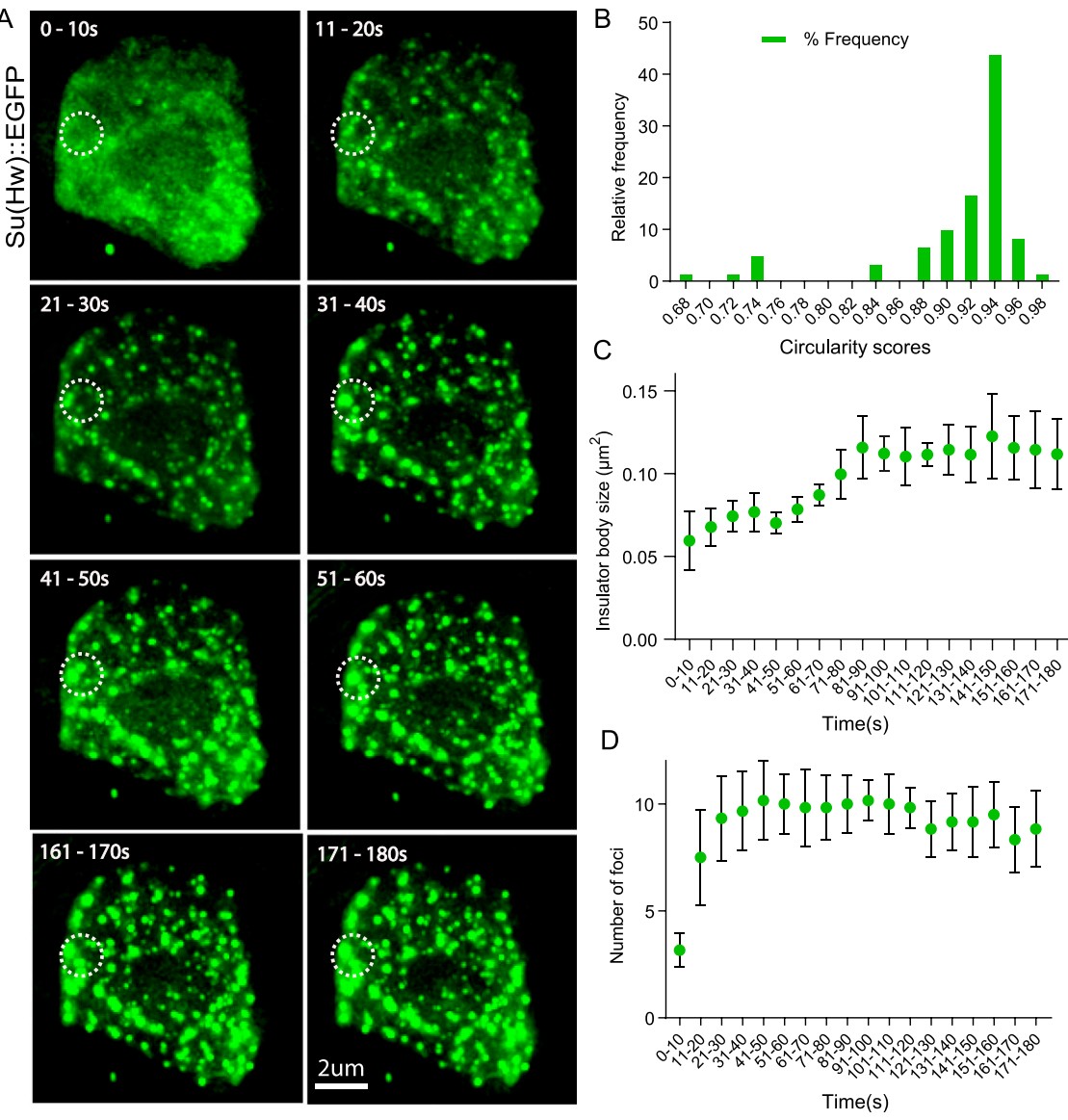

**Figure 3. Insulator bodies undergo fusions to form enlarged circular structures.**
**(A)** GFP is tagged to Su(Hw) protein labeled as Su(Hw)::EGFP. Images are taken every 10 s for 3 min. The first six (0–60 s) and last two (161–180 s) recordings are being shown. At 0–10 s, insulator-binding proteins are uniformly distributed. Insulator-binding proteins start forming speckles from around 10–20 s. As stress conditions prolong, the speckles start to fuse into larger bodies. An example of such fusion is shown in the white circles. **(B)** Most of the insulator bodies at the final time of 180 s exhibit perfect spherical structures using a scale of 0–1 for least circular to perfect circularity, respectively. **(C)** Insulator bodies increase in size steadily and then roughly plateau afterward. **(D)** Insulator bodies increase in number steadily and then roughly plateau afterward.

instance comprises more than 4,500 unique proteins (Ahmad et al, 2009), whereas stress granules contain over 300 proteins and more than 1,000 RNA transcripts (Markmiller et al, 2018). These constituents are intimately linked to the biological functions of the MLOs. However, the full complement of biomolecules in the insulator bodies is yet to be ascertained. Independent reports suggest that insulator bodies comprise of many unrelated proteins including the EAST protein (Melnikova et al, 2019), the gypsy insulator complex proteins, BEAF32, dCTCF (Schoborg et al, 2013), and the phosphorylated histone variant γH2Av (submitted). Interestingly, the mammalian CTCF and cohesin subunits also form clusters of characteristic size of ~200 nm (Hansen et al, 2019, 2020). However,

there has not been a demonstration of cohesin clustering with IBPs in *Drosophila*. Given the intimate role played between cohesin and the mammalian CTCF insulator in genome organization, we asked whether cohesin associates with insulator bodies by using a Rad21: myc fusion expressed under the *tubulin* promoter and an anti-Myc antibody (UBPBio).

We used polytene chromosomes to co-immunostain Rad21::Myc with the CP190 insulator protein. Results show that an important fraction of Rad21::Myc sites colocalizes with CP190 (Fig 6A and B). These results coincide with published data from ChIP (Van Bortle et al, 2014) and support the notion that cohesin is enriched at IBPs both in diploid cells and in polytene chromosomes (Stow et al,

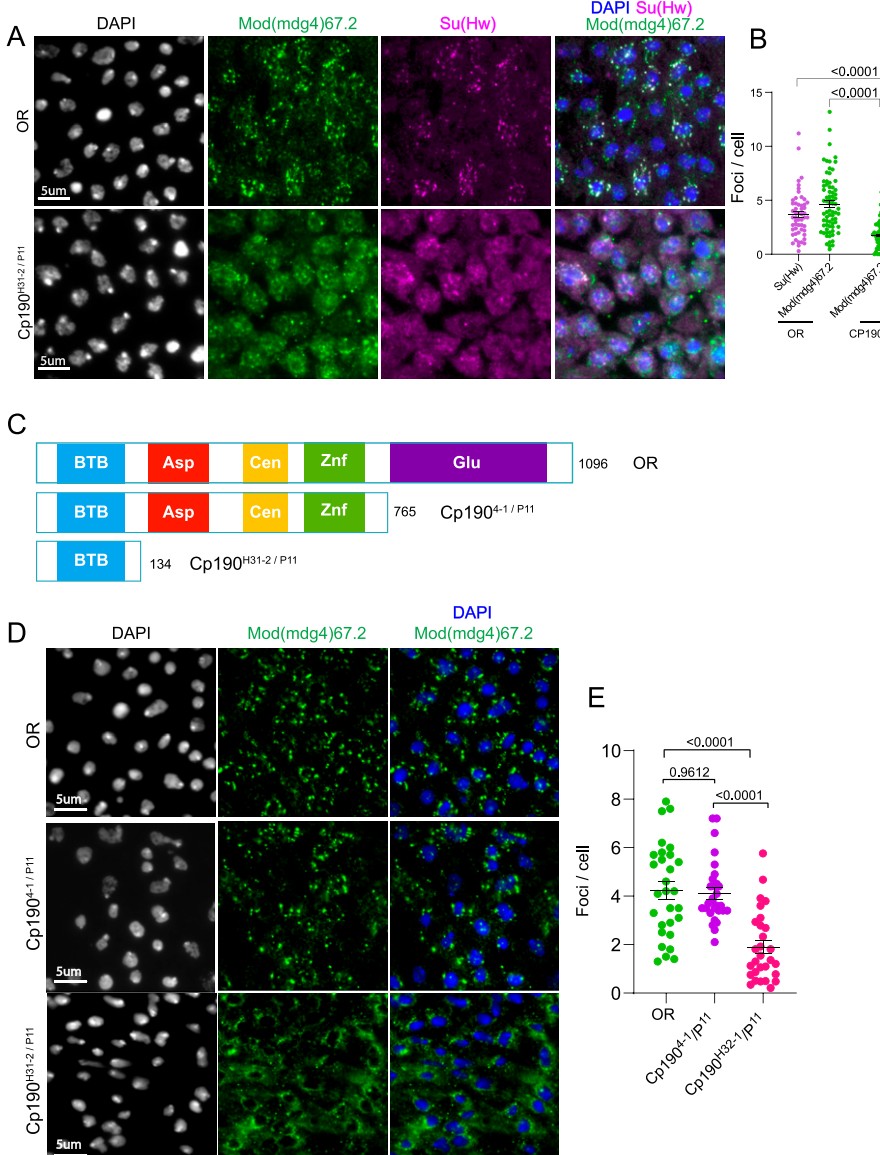

**Figure 4. Insulator bodies exhibit scaffold–client properties.**
**(A)** Investigating Cp190's role in insulator body formation. In the OR panel, both Mod(mdg4)67.2 (green) and Su(Hw) (magenta) display high number of insulator foci compared with the respective foci formed in the *cp190* trans-homozygote mutant background (Cp190[H31-2/P11]). **(B)** Quantitative comparison of the insulator body number formed by Mod(mdg4)67.2 and Su(Hw) in wild-type (OR) and *cp190* mutant backgrounds. In the *cp190* mutant background, the numbers of insulator bodies by Mod(mdg4)67.2 and Su(Hw) are significantly lower than those in the OR. **(C)** Top panel: wildtype (OR) CP190 protein displaying its five domains, BTB, aspartic-rich (Asp), centrosome-binding domain (Cen), zinc finger domain (Znf), and glutamic acid–rich domain. Middle panel: trans-heterozygote *cp190* mutant (Cp190[4-1/P11]) showing removal of only the glutamic-rich amino acid. Bottom panel: trans-heterozygote *cp190* mutant that shows removal of all four non-BTB domains (Cp190[H31-2/P11]). **(D)** Quantitative comparison of the number of insulator bodies between wild-type OR and the non-BTB domain mutants Cp190[4-1/P11] and Cp190[H31-2/P11]. **(E)** Ordinary one-way ANOVA followed by Tukey's multiple comparisons test showing a significantly lower number of insulator bodies between OR and the non-BTB mutant Cp190[H31-2/P11] ($P$-value < 0.0001) but not for the glutamic-rich domain mutant Cp190[4-1/P11] ($P$-value = 0.9612). For each genotype, three correlated biological replicates were combined. $P$-values < 0.05 are deemed significant.

2022). Next, we analyzed imaginal disc cells expressing Rad21::Myc under osmotic stress conditions and performed immunostaining using anti-Myc and anti-CP190 antibodies. Results show that, Rad21::Myc overlaps substantially with CP190 foci (Fig 6C and D). This implies that insulator bodies are not just IBPs but consist of a repertoire of other proteins involved in genome organization, including cohesin.

## Phosphorylation of H2Av modulates insulator body formation

Posttranslational modifications including phosphorylation, SUMOylation, and methylation are documented to alter the multivalency of proteins and are therefore prominent modulators of condensation responses (Hofweber & Dormann, 2019; Owen & Shewmaker, 2019). For example the assembly of stress granules relies on the phosphorylation of G3BP and PABP (Rai et al, 2018),

and purified human heterochromatin protein 1α (HP1α) undergoes LLPS in a phosphorylation-dependent manner (Larson & Narlikar, 2018). Interestingly, we observed that the DNA damage marker γH2Av and not its unphosphorylated form, H2Av, is a positive regulator of insulator body formation. We inferred that H2Av phosphorylation contributes to the multivalent interactions required for the assembly of insulator bodies. We therefore sought to investigate how H2Av phosphorylation affects the formation of insulator body condensates. To this end, we tested the effect of phosphatase inhibition on the number of stress-induced insulator bodies. We generated insulator bodies in the presence of 50 nM okadaic acid and detected insulator bodies by fluorescence microscopy using an antibody against Cp190. The number of insulator bodies was calculated and compared with a control sample. Okadaic acid is a potent inhibitor of serine/threonine phosphatases PP1 and PP2A (Bialojan & Takai, 1988; Cohen et al, 1990).

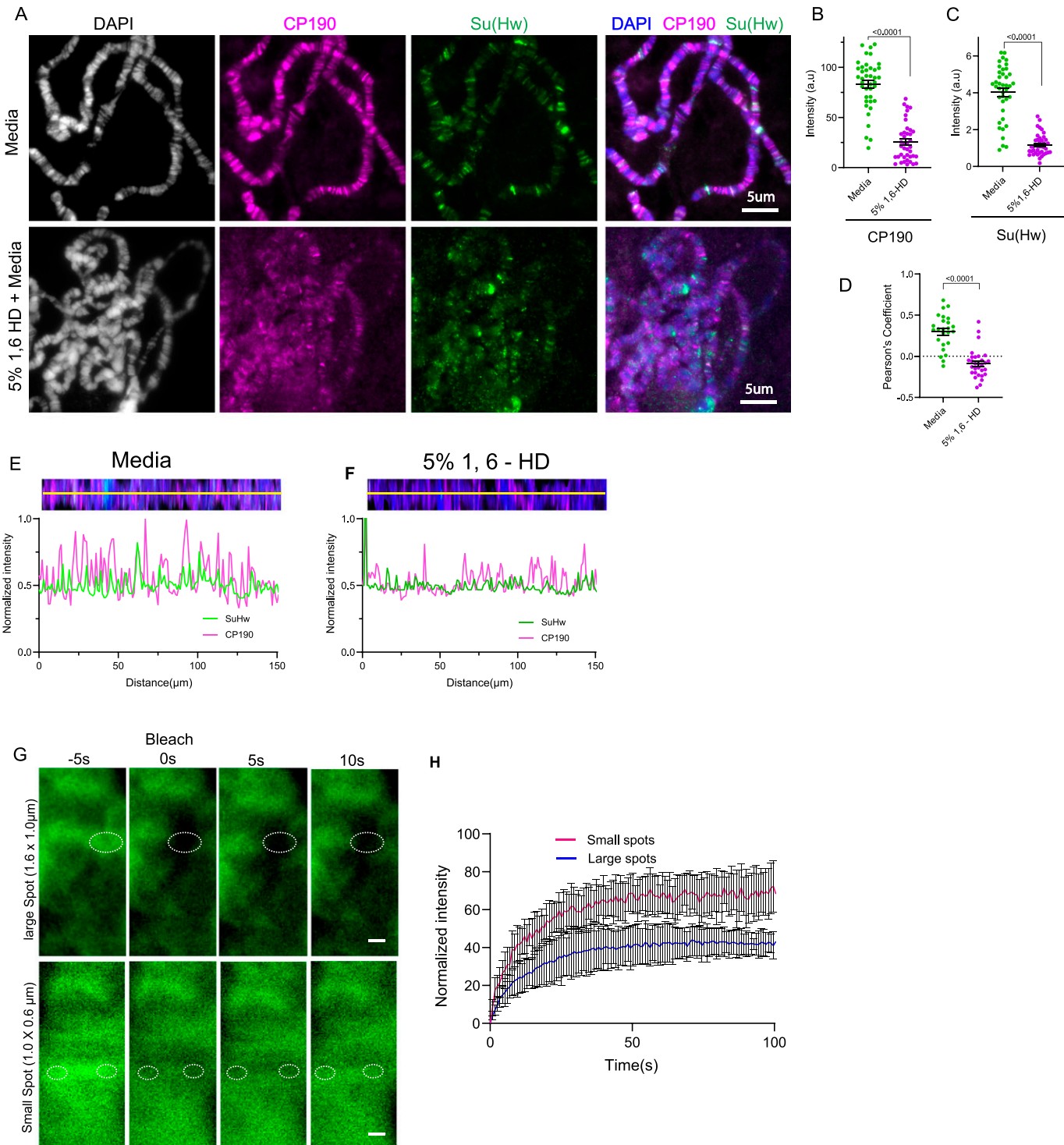

**Figure 5. Insulator proteins possess liquid–liquid phase separation features at physiological conditions on polytene chromosomes.**
**(A)** Constitutive bindings of insulator proteins are sensitive to 1,6-hexanediol depicted by loss of Cp190 and Su(Hw) bands in 5% 1,6-hexanediol treated polytenes compared with the media treated ones. Polytene chromosomes are stained with CP190 (magenta), Su(Hw) (green), and DAPI (gray and blue). **(B, C)** Quantitative measurement of Cp190 intensity (B) and Su(Hw) (C) on polytene using DAPI as region of interests (ROI). For each treatment, three biological replicates were combined. **(D)** Pearson's correlation coefficient (PCC) for Cp190 signal with Su(Hw) signal is plotted, with each point representing the polytene genome of each cell. Overlaps between CP190 and Su(Hw) measured with Pearson's correlation coefficient are significantly reduced with increasing 1,6-hexanediol concentration. **(E, F)** Normalized intensity of Su(Hw) and Cp190 channels plotted against distance along the yellow line scans in media (E) and 5% 1,6-HD (F). **(A)** The polytene chromosomes from (A) were stretched into single linear strand before the line scans. **(G)** Fluorescence recovery after photobleaching (FRAP) comparison between large (1.6 × 1.0 $\mu m$) and small (0.6 ×

The low concentration was to prevent any effect of okadaic acid on the structural integrity of the cells and to increase the specificity for PP2A (Ferron et al, 2014; Fu et al, 2019). PP2A in turn dephosphorylates γH2Av (Nakada et al, 2008). We found that inhibition of γH2Av dephosphorylation significantly decreased the number of insulator bodies per cell (Figure 7A and B). These results highlight an involvement of kinase activity in insulator body formation. In summary, phosphorylation of H2Av modulates the LLPS process of insulator proteins and may contribute to the material properties of insulator bodies.

## Discussion

A plethora of eukaryotic biological processes including stress response and gene transcription are regulated in part through the formation of biomolecular condensates (Ulianov et al, 2016; Stam et al, 2019; Sanders et al, 2020). In particular, the role of membraneless organelles in 3D genome organization is spurring numerous research efforts, owing to their preferential interactions with specific chromatin regions (Weierich et al, 2003; Zhang et al, 2004; Quinodoz et al, 2018). It is becoming increasingly clear that the biophysical process of LLPS underlies the formation of these membraneless organelles. We have previously reported that insulator bodies formed from *Drosophila* chromatin insulator proteins are dynamic salt-stress response bodies with recovery half-times in the order of seconds (4–15 s) (Schoborg et al, 2013). However, to our knowledge, whether insulator bodies are formed through liquid phase separation has not been explored. Here, we have provided evidence supporting that *Drosophila* insulator proteins possess LLPS properties and that other chromatin architecture proteins such as cohesin and γH2Av also contribute to the formation of insulator bodies. We propose a model by which the contribution of these proteins to the 3D organization of the genome is mediated at least in part by liquid phase separation.

First, we show that known constituents of insulator bodies have high intrinsic disorder tendency, a property shared by most protein components of MLOs. The broad classification of proteins based on disorder includes structured proteins (0–10% disorder), moderately disordered proteins (10–30% disorder), and highly disordered proteins (30–100% disorder) (Gsponer et al, 2008; Edwards et al, 2009; Van Der Lee et al, 2014). We found that the disorder levels displayed by IBPs fall within the highly disordered protein category. In addition, IBPs reveal a high density of charged residue tracts but low levels of kink-forming aromatic residues (Figs 1A and B and S2), indicating a likelihood of electrostatic-mediated clustering of insulator proteins (Fig 1A and B). Unlike stretches of residues in which charges are uniformly dispersed, tracts of contiguous charged residues are thought to provide weak electrostatic forces that contribute to phase separation (Somjee et al, 2020). The importance of such electrostatic interactions has been observed, among others, in the IDRs of histone H1 (Turner et al, 2018), nucleophosmin (Mitrea

et al, 2018), Ddx4 (Nott et al, 2015), and CBX2 (Plys et al, 2019), which form the well-characterized condensates of histone locus bodies, the nucleolus, germ granules, and polycomb bodies, respectively.

Drawing inspiration from polymer physics, IDRs are described either as polyampholytes or polyelectrolytes based on the patterning of their charged residues, allowing the prediction of their conformational ensembles (Das & Pappu, 2013; Das et al, 2015; Holehouse et al, 2017; Bianchi et al, 2020). Interestingly, we found that based on this classification the known insulator body components such as Su(Hw), Mod(mdg4)67.2, Cp190, dCTCF, and γH2Av fit with the "Janus Sequence" IDR classification (Fig 1D). Proteins within this group display both weak and strong polyampholyte features enabling them to either collapse or expand, depending on the environmental conditions (Das & Pappu, 2013). This context dependency may explain why insulator proteins coalesce into bodies during salt stress that dissolve when isosmotic conditions are restored (Schoborg et al, 2013).

The functional implications of these unique IDR features of insulator proteins are not yet well understood. However, previous studies indicated an abrogation of insulator enhancer-blocking function upon the removal of the C-terminal glutamate-rich and the glutamine-rich domains of Cp190 and Mod(mdg4)67.2, respectively (Golovnin et al, 2008; Oliver et al, 2010). Interestingly, the glutamic acid–rich region of Cp190 is also required for its dissociation from chromosomes during heat-shock (Oliver et al, 2010). Though these results do not decouple insulator body effect of the truncated Cp190 domains from effect of the charged residues, the results emphasize the importance of the charged residues in the LLPS properties of IBPs. However, the high PScores (Fig S2) by the IBPs raise the possibility of other forces including hydrophobic, π–π, and cation–π interactions as contributing forces in insulator body formation. Indeed, 1,6-hexanediol which dissolves phase separation assemblies by disrupting weak hydrophobic protein–protein or protein–RNA interactions (Kroschwald et al, 2017; Itoh et al, 2021) dissolved insulator bodies (Fig 2A–C). Coupled with the low LARKS and the high electrostatic properties mentioned above, the sensitivity of insulator bodies to 1,6-hexanediol implies that there is a contribution of both hydrophobic and electrostatic forces in their formation and maintenance. Although LLPS condensates such as P bodies are sensitive to this alcohol, solid-like condensates such as protein aggregates and cytoskeletal assemblies are not (Wheeler et al, 2016). Our data are consistent with the notion that insulator bodies are liquid droplets and not solid aggregates.

The fusion and relaxation ability of condensates into spherical structures are important qualitative proxies for LLPS (Hyman et al, 2014; Alberti et al, 2019). Interestingly, we demonstrated a predominantly spherical and fusion behavior of insulator bodies (Fig 3). It is argued that the spherical nature of LLPS-mediated condensates is a reflection of a change in refractive index and surface tension that arise from formation of a distinct phase separated from the surrounding nucleoplasm (Hyman et al, 2014; Chong et al, 2018). On the other hand, the fusion behavior maybe a consequence of an enrichment inhibition, whereby certain mechanisms

---

1.0 μm) spots on Su(Hw)-GFP–tagged bands on polytene chromosomes. **(H)** A plot of recovery (normalized intensity) and time after photobleaching of large and small spots on Su(Hw)-GFP–tagged bands on polytene chromosomes.

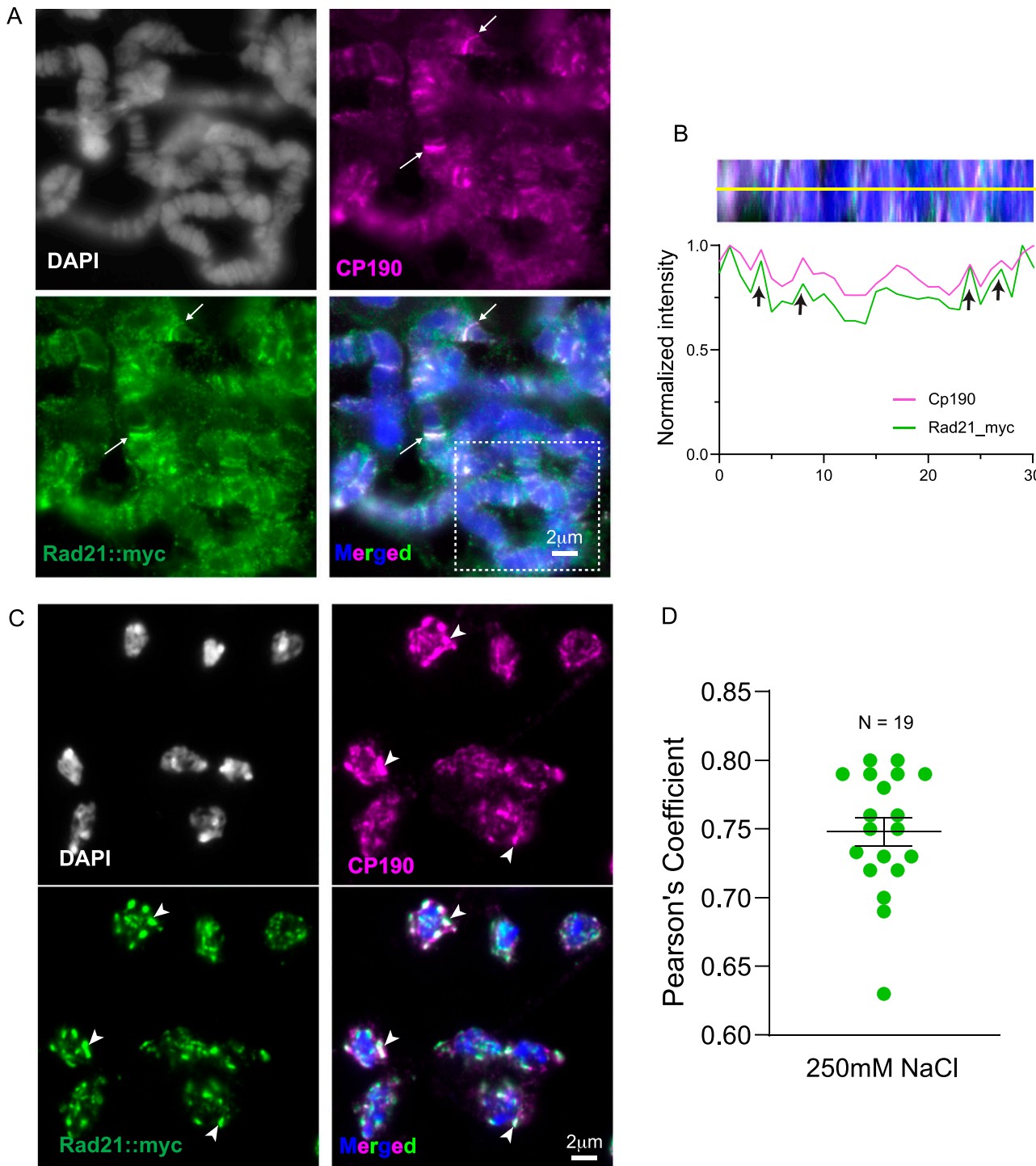

**Figure 6. The cohesin subunit RAD21 colocalizes with CP190 on polytene chromosomes and forms liquid–liquid phase separation condensates with insulator proteins.**
**(A)** Polytene chromosomes immunostained with Rad21::myc and Cp190. White arrows show examples of regions of high colocalization between Rad21::myc and Cp190 on the polytene chromosomes. **(A, B)** Inset (white broken line square) from merged figure in (A) is stretched into a linear strand. Black arrows show regions of high overlap between Rad21 and Cp190. **(C)** myc-tagged Rad21 (Rad21::myc) associates with insulator bodies formed in wing imaginal disc cells. White arrows show examples of regions of high colocalization between Rad21::myc and Cp190 in insulator bodies. **(D)** Pearson correlation (PCC) between CP190 and Su(Hw) showing high overlap between Rad21::myc and Cp190 in insulator bodies.

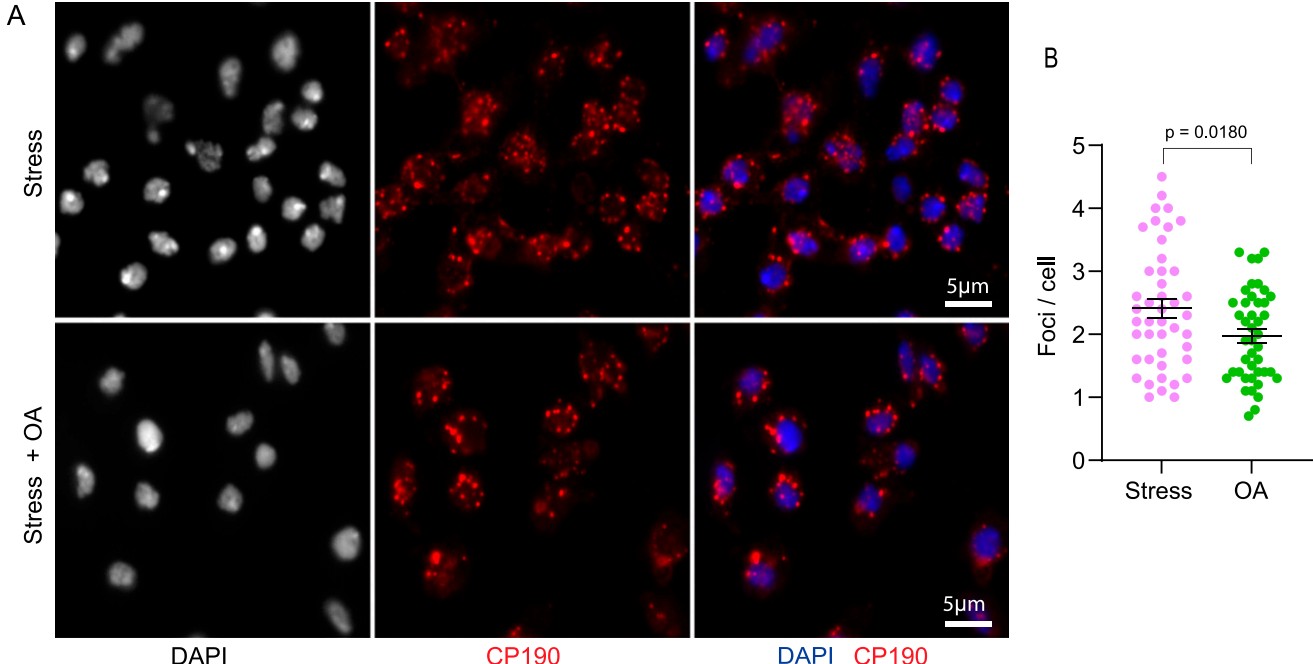

**Figure 7. Phosphorylation of H2Av modulates insulator body formation.**
**(A)** Insulator bodies generated in 250 mM NaCl (stress) with okadaic acid (bottom panel) and without okadaic acid (top panel). **(B)** Quantitative comparison of insulator bodies per cell between stressed and okadaic acid showing a significantly lower number of bodies in the presence of okadaic acid (*P*-value = 0.0180). For each treatment, three biological replicates were combined.

including posttranslational modifications exist to limit the size of larger condensates, allowing the coexistence of multiple ones (Söding et al, 2020).

Importantly, the fusion of small insulator bodies into larger ones roughly plateaued with time, emphasizing a likelihood that the number and sizes of insulator bodies scale with concentration of its constituents. The concentration dependence of LLPS-mediated bodies is typically delineated with phase diagrams where two conditions, for example, protein concentration and salt are systematically changed to determine in which conditions a dense phase is detectable (Banani et al, 2017; Alberti et al, 2019). Although such optimum conditions have not been established for insulator bodies, larger insulator bodies have been recorded at concentrations below 250 mM NaCl (Schoborg et al, 2013), implying that insulator proteins' phase separation is sensitive to ionic concentration and that it can occur in physiologically relevant contexts. We tested this possibility by incubating non-stressed polytene chromosomes with 1,6-hexanediol. Consistently, insulator proteins are not only sensitive to 1,6-hexanediol in their salt stress-induced bodies but also in their cognate DNA-associated form on polytene chromosomes (Fig 5A–F).

Therefore, insulator proteins may not just participate in the formation of stress response condensates but may also form constitutive assemblies of ribonucleoproteins during normal physiological conditions. In fact, others have argued the existence of two forms of chromatin insulator condensates; the hyperosmotic stress-induced bodies and the constitutively refined speckles relevant for long distance genomic site interactions including contacts between distant Hox loci in *Drosophila*; a phenomenon

known as Hox-gene kissing (Buxa et al, 2016). Moreover, a study in human cells indicated a partial compromise in the 3D genome through suppression of LLPS by 1,6-hexanediol (Ulianov et al, 2021) and the chromatin architecture proteins CTCF and SMC3 exhibited moderate sensitivity to 1,6-hexanediol elsewhere (Shi et al, 2021). These highlight a possible conservation and relevance of constitutive phase separation properties of genome architecture proteins across species.

The LLPS-mediated constitutive assembly of insulator proteins is buttressed by the dependence of the recovery of photobleached Su(Hw) polytene chromosome bands on sizes of the bleached area (Fig 5G and H). The size-dependence of polytene band recovery highlights the contribution of not just binding but diffusion in the insulator proteins interactions with the chromatin as explained elsewhere (Sprague & McNally, 2005; McSwiggen et al, 2019b). Taken together, these results suggest that insulator proteins possess inherent LLPS abilities that may confer unique functions on their various continuums of assemblies, including insulator speckles under normal conditions, and stress-induced insulator bodies during osmotic stress.

We previously demonstrated the reliance of insulator bodies on the phosphorylation of the *Drosophila* histone variant H2Av (submitted). In this work, inhibition of dephosphorylation significantly decreased the number of insulator bodies (Fig 7A and B). This may be because of an impact of the phosphorylation on the rheology or material property of the bodies. It is therefore likely that both phase separation-enhancing kinase and a condensate-dissolving phosphatase exist for the modulation of insulator bodies as seen in other membraneless organelles including stress

granules (Rai et al, 2018), transcriptional condensates (Guo et al, 2019), and P-bodies (Luo et al, 2018). An important question that remains unanswered is whether kinase and phosphatase activity also modulate the insulator body activity and therefore insulator activity at IBP sites in the genome.

Our data also suggest that insulator bodies follow a scaffold–client model in that two of their components, Cp190 and Mod(mdg4) 67.2, appear to be crucial for their formation (Figs 4 and S5), whereas Su(Hw) serves as a "client" protein. Cp190 and Mod(mdg4)67.2 may thus be essential scaffolds with others like Su(Hw) serving a regulatory function. This is surprising judging that unlike Su(Hw), both Mod(mdg4)67.2 (Büchner et al, 2000) and Cp190 (Pai et al, 2004) are physically and functionally connected to insulators without binding directly to DNA. Although previous studies suggested dependence of DNA sites in insulator protein assembly (Gerasimova & Corces, 1998; Gerasimova et al, 2000; Ghosh et al, 2001), it has recently been suggested that insulator bodies are formed at chromatin free regions of the nucleus (Schoborg et al, 2013), signifying that insulator proteins may not rely on DNA as a polymer to form condensates. The veracity of any of these arguments is important because a distinction has been made between LLPS and bridging induced polymer–polymer phase separation (PPPS) based on the dependence of the length or abundance of DNA or RNA polymer scaffolds (Brackley et al, 2013; Brackley & Marenduzzo, 2020; Ryu et al, 2021). Remarkably, the proposed client Su(Hw) has both the lowest disorder tendency and PScore but higher LARK segments than CP190. Although this somehow gives credence to the scaffold function of CP190 and Mod(mdg4)67.2, it also explains the reliance of the charged residues and not kinked segment formation from amino acids with π-contacts. Further studies would be required to differentiate LLPS from PPPS properties of insulator proteins. However, findings from this work shows insulator bodies possess more of LLPS features than they would for PPPS. The reliance of insulator bodies on Cp190 in particular is intriguing as all *Drosophila* insulator protein complexes contain Cp190 and is also highly enriched at TAD borders (Bag et al, 2021).

The presence of the cohesin subunit Rad21 in insulator bodies (Fig 6C and D) highlights a possible concerted function of cohesin and insulator proteins in *Drosophila*, similar to their synergistic genome organization role in mammals through the loop extrusion model (Brandão et al, 2018; Costantino et al, 2020). A recent study showed that the yeast cohesin exhibits pronounced clustering on DNA, with all the hallmarks of biomolecular condensation (Ryu et al, 2021). Interestingly, both mammalian CTCF (Zirkel et al, 2018) and *Drosophila* insulator proteins (Schoborg et al, 2013) undergo cell death–induced clustering. These give further credence to conserved LLPS-induced genome organization roles of genome architecture proteins. Similar to the roles of cohesin and the CTCF insulator in human genome organization, these results highlight an important insulator-cohesin combined effect in the organization of *Drosophila* genome.

In conclusion, in this work, we show insulator proteins possess LLPS properties that allow a stimulus response and the constitutive formation of biomolecular condensates. We ascribe this to the contribution of both electrostatic and hydrophobic forces, owing to the possession of oppositely charged "blocks" of residues and sensitivity to 1,6-hexanediol, respectively. In addition, we have demonstrated that beside core insulator proteins, key components of insulator bodies include cohesin and the *Drosophila* histone variant γH2Av. Although the enhancer-blocking and 3D-genome organization roles of insulator bodies remain controversial, the exploration of these LLPS properties will help to address the gap in knowledge of the biological function of insulator bodies in future work.

# Materials and Methods

### Fly stocks and husbandry

All stocks were maintained on a standard cornmeal agar fly food medium supplemented with yeast at 20°C; crosses were carried out at 25°C. Oregon R was used as the wild-type stock. The stocks *cp190^{H31-2}*/TM6B, *cp190^{P11}*/TM6B, *w^{1118}*;*su(Hw)^V*/TM6B, and *mod(mdg4)^{u1}*/TM6B Tb1 are maintained in our lab and were originally obtained from Victor Corces (Emory University). Our laboratory generated the su(Hw)::eGFP line used for the FRAP experiment. Microinjection to generate transgenic lines yw; P{SuHw::EGFP, w+} was performed by GenetiVision. The eGFP was expressed by crossing the yw; P{SuHw::EGFP, w+} to w*; vg-Gal4; TM2/TM6B line. We obtained the w1118; PBac(RB)su(Hw)e04061/TM6B, Tb1 stock from the Bloomington Drosophila stock center (BDSC: 18224). The su(Hw)^{e04061} mutant allele contains an insertion of a piggyBac transposon in the 5′ end of the second exon of *su(Hw)* (Baxley et al, 2011; Schoborg et al, 2013), whereas the *su(Hw)^V* carries a deletion of the *su(Hw)* promoter (Harrison et al, 1992). The line *w;vtd;Tub*>Rad21-TEV-myc is a gift from the McKee Lab, University of Tennessee, Knoxville, and was originally obtained from the Bloomington Drosophila stock center (RRID:BDSC_27614). *w;vtd;Tub*>Rad21-TEV-myc expresses myc-tagged vtd (Rad21) protein in all cells under control of the alphaTub84B promoter (Pauli et al, 2008).

### Antibodies

Rabbit polyclonal IgG antibodies against Su(Hw), Mod(mdg4)67.2, and CP190 and rat polyclonal IgG antibody against Su(Hw) were previously generated by our lab (Wallace et al, 2010; Schoborg et al, 2013). Mouse monoclonal antibodies against the phosphorylated form of H2Av (Lake et al, 2013) were obtained from the Developmental Studies Hybridoma Bank, created by the NICHD of the NIH and maintained at the University of Iowa, Department of Biology, Iowa City, IA 52242. Polyclonal rabbit antibodies against H2Av were purchased from Active Motif (RRID:AB2793318). The monoclonal Mouse antibody anti-myc was used to detect Rad21::myc and was obtained from Ubiquitin-Proteasome Biotechnologies (#Y1090; UBPBio). All primary and secondary antibodies were diluted 1:1 in glycerol (BP229-1, lot 020133; Thermo Fisher Scientific) and used at a final dilution of 1:200. The following secondary antibodies were used in this study: Alexa Fluor 594 goat anti-rabbit (A-111037, lot 2079421; Invitrogen), Alexa Fluor 488 donkey anti-rabbit (lot 1834802, A-21206; Invitrogen), Alexa Fluor 488 goat anti-guinea pig (lot 84E1-1, A-11073; Invitrogen), Texas red donkey anti-rat (712-075-150; Jackson Immuno-Research Laboratories), and Alexa Fluor 488 goat anti-mouse (lot 1858182, A-11001; Invitrogen).

**Life Science Alliance**

### Stress treatment and immunostaining of larval tissues

PBS was used as a physiological media. Osmotic stress was induced using PBS supplemented to 250 mM NaCl. Wing imaginal discs were dissected from wandering third instar larvae in PBS. To induce osmotic stress, the media were removed and quickly replaced with PBS::250 mM NaCl for 30 min as previously described (Schoborg et al, 2013). Control tissues were kept in PBS for the same incubation time. Tissues were then placed into fixative prepared from 50% glacial acetic acid (A38-212, lot 172788; Thermo Fisher Scientific) and 4% para-formaldehyde (43368, lot N13E011; Alfa Aesar). For polytene chromosomes, squashes were prepared by lowering a slide on top of the sample then turning it over, placing it between sheets of blotting paper, and hitting the coverslip firmly with a small rubber mallet. Same procedure was followed for both wing disc cells except the slides were pressed firmly against a hard platform with the rubber mallet rather than directly hitting the slides. Slides were also cryo-fixed by dipping in liquid nitrogen, and coverslips were then removed, and samples were incubated in blocking solution (3% powdered nonfat milk in PBS + 0.1% IGEPAL CA-630) (18896, lot 1043; Sigma-Aldrich) for 10 min minimum at room temperature. Slides for both wing discs and polytene chromosomes were then incubated with primary antibodies at 4°C overnight in a humidifying chamber.

After overnight incubation, the slides were washed three times in PBS containing 0.1% IGEPAL CA-630 followed by a 3-h incubation with secondary antibodies in the dark at room temperature. The washing step with PBS and 0.1% IGEPAL CA-630 was then repeated, and the slides were treated with DAPI solution of 0.5 $\mu$g/ml (D1306; Thermo Fisher Scientific) for 1 min followed by one more time washing in PBS alone. Mounting was done with VECTASHIELD Antifade Mounting Medium (lot ZF0409, H-1000; Vector Laboratories). The coverslips were then sealed with clear nail polish.

### Fluorescence and confocal microcopy

All microscopy for immunostaining was performed on a wide-field epi-fluorescent microscope (DM6000 B; Leica Microsystems) equipped with a 100×/1.35 NA oil immersion objective and a charge-coupled device camera (ORCA-ER; Hamamatsu Photonics). Simple PCI (v6.6; Hamamatsu Photonics) was used for image acquisition. FIJI, an open source image-processing package based on ImageJ2 was used for image analysis (Schindelin et al, 2012). All contrast adjustments are linear. Images were further processed in Adobe Photoshop CS5 Extended version 12.0 ×64 and then assembled with Adobe Illustrator CS5, version 15.0.0. Python version 3.7 and GraphPad Prism version 9.0.0 (224) (GraphPad Software) were used to perform the statistical analyses. Only most typical cases of cytological localizations are shown on the figures in the manuscript in the "Results" section. However, the conclusions are drawn on the basis of analysis of large numbers of polytene nuclei and wing disc cells collected in triplicates.

Insulator body number and sizes were analyzed using Particle Analysis feature in ImageJ software with a lower size limit of diameter = 0.2 $\mu$m and upper size limit of diameter = 1 $\mu$m. The circularity index of insulator bodies was calculated by $4\pi A/C^2$ where A is the area of the insulator body mask and C is the perimeter of the insulator body mask. These calculations were done with FIJI. Circularity value of 1.0 indicates a perfect circle and an approach toward 0.0 as an increasingly elongated polygon.

FRAP experiments on polytene chromosomes were done with Leica SP8 confocal microscope at the Advanced Microscopy and Imaging Center of University of Tennessee, Knoxville. Briefly, third instar larvae polytene chromosomes expressing Su(Hw)-EGFP were dissected and immediately immersed in PBS. Two oval (1.6 × 1.0 $\mu$m) and (0.6 × 1.0 $\mu$m) ROI spots were selected on the Su(Hw)::EGFP bands and were bleached simultaneously using an argon laser set to 80% (50 mW) at "Zoom in" mode. Low laser intensity was set for fluorescence imaging pre- and post-bleaching. Frames were acquired every second. The GFP recoveries were recorded and monitored in real time using Leica Acquisition System and terminated once the curve plateaued. Raw intensities were corrected for photobleaching and subtracted from background and normalized with the final prebleach frame intensity taken to be 1. Recovery curves were plotted and fitted to a one-phase association exponential function using Prism 9 software (GraphPad Software).

### 1,6-Hexanediol treatment

1,6-Hexanediol was obtained from Sigma-Aldrich (Cat. no. 240117). 5% 1,6-hexanediol was prepared with PBS, or PBS:250 mM NaCl were used. PBS served as media for all the experiments. To check for the effect of 1,6-hexanediol on insulator bodies, the cells were first stressed with 250 mM NaCl prepared from PBS and then quickly replaced with the hexanediol solution prepared with 250 mM NaCl for 2 min. For the effect of the alcohol on IBP intensities on polytene chromosomes, the media was removed and quickly replaced with the hexanediol solution prepared with PBS for 2 min.

### Analysis of protein disorder, charge, and LLPS predictions

Disorder tendency for individual insulator proteins was calculated using the IUPred2 algorithm (Mészáros et al, 2018). The number of disordered regions, number of disordered residues, average prediction score, and the overall percent disorder for the proteins were derived from the Predictors of Natural Disordered Regions algorithm (Peng et al, 2006). Various properties of the disordered regions in insulator proteins including the $\kappa$, NCPR, and FCR were calculated from the webserver Classification of Intrinsically Disordered Ensemble Regions (CIDER) (Holehouse et al, 2017). A window size of 20 residues (blob index) was used to plot the NCPR graphs. The low-complexity aromatic-rich kinked segments (LARKS) were determined from the webserver, LARKSdb (Hughes et al, 2018). The number of LARK segments was counted in binary: either a segment is predicted to form a LARKS or it is not irrespective of length of the bars. The prediction of potential phase separation proteins (PSPs) and calculations of PScores were done with the webserver PSPredictor (Vernon et al, 2018).

### Fluorescence intensity and colocalization analysis

Quantitation of fluorescent images was performed using ImageJ. For quantitation of signal in individual nuclei, nuclear boundaries were identified via thresholding and water shedding.

Amount of each protein in the images (i.e., the intensity of each channel) was analyzed using a macro script in FIJI (Schindelin et al, 2012). First, non-biased ROIs for each cell were generated automatically with the DAPI channel. A rolling-ball background subtraction algorithm was used for all images. Intensity measurements were made using the measure function. Numerous images of polytene and wing imaginal discs were collected. Within each experiment, all acquisition parameters were kept constant between slides. The Coloc2 plugin in FIJI was used for the colocalization measurements. This analysis is based on the Costes method (Costes et al, 2004) to determine appropriate thresholds for each channel. The colocalization results are reported using Pearson's correlation coefficient (PCC), ranging from +1 for perfect correlation and –1 for perfect anticorrelation (Dunn et al, 2011).

# Supplementary Information

# Acknowledgements

We are grateful to James R Simmons for the generation of FIJI macros used in the analysis of the various fluorescence microscopy images. We would like to thank the microscopy center manager at the University of Tennessee division of biology Jaydeep Kolape for his immense help in using the confocal microscopy, especially in the FRAP experiment. We appreciate the McKee lab for the supply of the line w;vtd;Tub>Rad21-TEV-myc. S2 cell culture was obtained from the *Drosophila* Genomics Resource Center (NIH 2P40OD010949). We also appreciate the two anonymous reviewers of this paper for their critical reading and insightful contributions that grately improved the initial manuscript. This work was supported partially by the College of Arts and Sciences and the Department of Biochemistry and Cellular and Molecular Biology.

## Author Contributions

B Amankwaa: conceptualization, resources, data curation, formal analysis, validation, investigation, visualization, methodology, and writing—original draft, review, and editing.
T Schoborg: data curation and investigation.
M Labrador: conceptualization, resources, data curation, formal analysis, supervision, funding acquisition, investigation, visualization, methodology, project administration, and writing—original draft, review, and editing.

## Conflict of Interest Statement

The authors declare that they have no conflict of interest.

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
