## [Reviewer comments · Life Science Alliance]

Life Science Alliance

Drosophila insulator proteins exhibit in vivo liquid-liquid phase separation properties

Bright Amankwaa, Todd Schoborg, and Mariano Labrador

DOI: <https://doi.org/10.26508/lsa.202201536>

Corresponding author(s): Mariano Labrador, University of Tennessee at Knoxville

Review Timeline:

Submission Date:	2022-05-27
Editorial Decision:	2022-06-27
Revision Received:	2022-07-05
Accepted:	2022-07-06

Transaction Report:

Please note that the manuscript was reviewed at Review Commons and these reports were taken into account in the decision-making process at Life Science Alliance.

Full Revision

Manuscript number: RC-xx-xx

Corresponding author(s): Mariano, Labrador

[Please use this template only if the submitted manuscript should be considered by the affiliate journal as a full revision in response to the points raised by the reviewers.]

*If you wish to submit a preliminary revision with a revision plan, please use our "Revision Plan" template. **It is important to use the appropriate template to clearly inform the editors of your intentions.**]*

1. General Statements [optional]

This section is optional. Insert here any general statements you wish to make about the goal of the study or about the reviews.

*The manuscript sought to look at whether insulator body components under physiological and stress conditions can be categorized as condensates, or droplets resulting from the process of liquid-liquid phase separation (LLPS). We approached this by analyzing the core insulator body components, CP190, Su(Hw), and Mod(mdg4)^{67.2} against hallmarks associated with liquid-liquid phase separation condensates. Our results reveal a novel component of insulator bodies and found that these bodies exhibit significant signatures associated with liquid-liquid phase separation condensates. These included high disorder tendencies, scaffold-client dependent assembly, extensive fusion behavior, sphericity, and sensitivity to 1,6-hexanediol. *In silico*, genetics, FRAP, fluorescent and confocal microscopy are some general approaches employed. A major concern raised by reviewers was the high concentration usage of the LLPS-dissolving alcohol, 1,6-hexanediol and its usage as a mainstream test. We have hence limited its usage to two experiments and resorted to a much lower concentration.*

This section is mandatory. Please insert a point-by-point reply describing the revisions that were already carried out and included in the transferred manuscript.

All author responses are italicized. Reviewer comments are placed on top of the italicized author responses

REVIEWER 1

Full Revision

The hexanediol treatment is described in insufficient detail and seem to suffer from a few serious issues. Unless these details are supplied and the explained issues are addresses, the treatments cannot be considered technically reliable. The issues are the following. The methods section does not state how long the treatment by hexanediol proceeds in each specific experiment. Hexanediol has severe effects on overall cell activity and integrity, so short treatments are preferable (5 min or less), after which the media must be replaced to largely remove hexanediol. This must even be done if fixation of samples directly follows the hexanediol treatment, as the substance can further degrade cells during progressive fixation. Also, in the stress experiments, the stressor media is replaced by regular media with hexanediol - this is not a fair comparison, as the stressor is removed during the treatment. It is unclear whether dissolution of droplets is due to hexanediol or stressor removal. The percentages of hexanediol are also excessive. In the recent literature that addresses hexanediol side effects on chromatin state and transcription, concentrations of 5% are sufficient to completely compromise chromatin state, chromatin organization, as well as transcription. Importantly, these side-effects are not via the pathway of "droplet dissolution", but bona fide side effects. These side effects are significantly less severe, to the extent of being negligible, at concentrations of 2-3% of hexanediol. All these concerns are not theoretical, the referee's lab has attempted the same experiments, and found that higher concentrations or exposure times can have effects as severe as completely dissolving cellular material. It is a little bit surprising that the choice of 10% hexanediol was considered appropriate, as the study cites a number of papers that clearly indicate that this concentration is excessive. Unless these points are addressed, the results are not conclusive.

We agree with the reviewer that both concentration and exposure time to hexanediol are important parameters that have to be carefully considered in our experiments. We agree that 10% concentrations may be excessive and we have removed all the 10% treatments from the results in the paper. We have maintained those of the 5% treatments. In our experiments we were careful to minimally expose cells to hexanediol for just 2 mins. Given that insulator proteins have shown that they can relocate very rapidly from insulator bodies to chromatin or from chromatin to insulator bodies (within seconds) we were careful to fix tissues immediately after exposure of hexanediol. We monitored the global effect of hexanediol by measuring the size of the nuclei using DAPI staining as a reference. In both 5% and 10% hexanediol we did not observe significant change in nuclear size.

Further, the hexanediol treatment is taken as a main piece of evidence for the study - this is not commonly considered a good approach, as there is too much debate around hexanediol and its mechanism of action as well as side effects. It makes the concerns expressed here the more relevant. Where possible, hexanediol treatment is typically combined with more targeted, mutational or chemical perturbation approaches to the putatively phase-separating macromolecular species. If possible, this would greatly strengthen the credibility of the presented conclusions.

In our final version of the manuscript, following the suggestions of the reviewer, hexanediol usage is limited to two experiments (figure 2 and figure 5) in which we tested the response of insulator bodies and insulator polytene bands to hexanediol exposure. We added new experiments using a more targeted mutational approach by disrupting 'sticker' residues from

the Cp190 insulator protein. More than 80% of the negatively charged amino-acids in Cp190 is found in its non-BTB domains. We have shown in a new experiment (now included in the manuscript) that removal of these non-BTB domains leads to a significant reduction in number of insulator bodies. Removal of just the glutamic-acid rich region reduces the NCPR from -0.09 to +0.03. Removal of all the non-BTB domain reduces the NCPR to -0.02. Interestingly, a similar truncation of the glutamic-rich CP190 region that corresponds to a similar reduced NCPR has been shown elsewhere (Oliver et al., 2010, BMC) to affect CP190 insulator function. Admittedly though, these results cannot decouple effect from the truncated Cp190 domain from just the reduction in the NCPR. An unambiguous attribution of the reduction of insulator body number to lowered NCPR would warrant targeted shuffling of the charged residues which we hope to follow-up in subsequent work.

In vitro experiments, which are typically required to support the specific interactions proposed to underly the in-vivo droplet formation, are missing. If these were carried out in previous published work, I missed these references.

We agree that in vitro experiments demonstrating the ability of these proteins to condensate would further support the conclusions of our manuscript. However, such experiments would most likely require the purification of one of its indispensable components referred to as 'scaffolds'. A key part of this manuscript is to provide in vivo and in silico evidence that these proteins have LLPS properties and use these evidence to identify at least one such scaffold protein. We identified CP190 as one such indispensable scaffold protein and we are working to purify this protein and be able to address the in vitro LLPS properties in a future manuscript.

A quantification of circularity without comparison against reference objects (Figure 3) is not conclusive. Shape quantifiers such as solidity or circularity can brought arbitrarily close to the value 1.0 by factors such as blurry imaging, or presence of objects that are only a few pixels large (any object consisting of a few pixels will have circularity close to 1). What is needed is the application of the exact same analysis pipeline to alternative nuclear structures, labeled and imaged as similarly as possible to the insulator bodies. These structures should be chosen, ideally, as structures that are not spherical, so as to provide evidence that the method can detect non-spherical objects, in case they are in fact present in the nuclei of the sample. Such an approach would establish that, under the particular sample conditions, labeling approach, and imaging conditions the circularity assessment can be considered conclusive.

The reviewer's suggested approach to verifying sphericity of insulator body condensates is quite insightful. Interestingly, BEAF-32 does not de-mix with the rest of the insulator proteins, but rather forms an irregularly shaped halo around insulator bodies upon osmotic stress (as shown in Schoborg et al 2013, JCB). The experiment initially presented in our first version of our manuscript did not include BEAF-32 data, but the cells used for live imaging in that experiment were coexpressing Su(HW)::GFP and BEAF-32::mcherry and data was collected for both proteins. Due to the non-spherical nature of the BEAF-32 halo, we decided to use it as a control

by comparing its circularity to that of the insulator bodies. We also highlight that the Fiji script we used to calculate circularity has been utilized by a number previous LLPS papers including Pascal Ziltener et al., 2020, FEBs letters. Our results show that insulator bodies are indeed spherical whereas the circularity of BEAF-32 is significantly reduced compared to that of Su(Hw)

In addition to the short-term experiments in Figure 3, longer time-lapses, or data from cells kept in the stress conditions for longer would give critical insight into the further ripening process, and whether the condensates are growth-limited or not. The authors also discuss Ostwald ripening - and this should become more visible over longer time scales, when ripening by coalescence has mostly run its course, droplets become larger and thus more stationary, as well as further apart from one another.

We thank the reviewer for this suggestion. In fact, we had collected data for a much longer period but did not realize of the significance of considering a longer time in this analysis. We have examined now insulator body behavior during a much longer period of time. Based on the new analysis, we conclude that coalescence but not Ostwald ripening explains the evolution of number of condensates.

In Figure 4, for a lot of the displayed elements, it is unclear to mean how exactly to interpret them. In panel A, I am not sure from the display how contrast adjustment was handled. My impression is, auto-scaling was used, so that loss of intense foci boosts the intensity of the background, and gives what appears like very large structures that span the cytoplasm. That is not ideal, and will be confusing to readers. Also, I do not know how to interpret these data. While in panel B, it is clear what is indicated (genotype at the bottom with lines, label directly under the distributions), in panel C, I cannot figure it out. Please fix, and I have to skip these data as non-conclusive for the time being. The overall idea of this experiment, testing client vs. scaffold characteristics, however, is a great idea and makes complete sense in the scope of the study question.

Panel C figure 4 looked at the contribution of the individual insulator proteins to the sensitivity of 1,6-hexanediol and, to an extension, the material properties of insulator bodies. Our results show that all the tested constituents contribute equally to 1,6-hexanediol sensitivity without any significant changes in the number of insulator bodies, irrespective of the background tested. In Figure 4 the X-axis labels are the proteins being tested in the experiment, whereas the underlined texts are the wildtype (OR) and mutant background (cp190).

Auto scaling was never used to analyze figure 4 or any other image in the manuscript. Because insulator bodies frequently localize between the chromosomal territories and the nuclear envelope, they frequently give the impression that are cytoplasmic in nature. This is because no marker for nuclear envelope is used in these experiments. However, in previous work we have carefully determined the nuclear nature of these bodies by coimmunostaining insulator bodies with nuclear lamina markers. Our work demonstrated that insulator bodies are never cytoplasmic (see Schoborg et al 2013, JCB). To avoid this impression when looking at this data we have also now provided more representative images for Figure 4.

Based on the results of Figure 5, the most likely scenario is a combination of DNA-binding and liquid-phase type weak interactions. Speaking, in this scenario, of LLPS as the primary driver of condensate formation is inaccurate. Rather, DNA-binding (maybe transient) in combination with an LLPS-mechanism jointly lead to condensate formation. The descriptions in the main text relating to Figure 5 reflect a somewhat mismatched vocabulary (related to LLPS) and the phenomena that the authors clearly deduce from their data (LLPS & DNA-binding in combination). These descriptions might be more in line with current understanding of this DNA-assisted process of condensate formation upon reading the three surface condensation papers suggested above. In these works, there are additionally many citations to previous papers on protein-DNA interactions and their relation to phase separation. I recommend to read up here, and think that it will be fairly clear how to reword this part of the results.

The reviewer is right and his assertions agree with our proposition that the previous Figure 5 suggested a combination of DNA-binding and liquid-phase type weak interactions. We agree with the reviewer that the wording used was a little confusing and we have reworded the main text relating to Figure 5 to more clearly reflect that both DNA binding and liquid-phase weak interactions are likely involved in the association of insulator proteins with chromatin.

I do not understand how a treatment with 5% and 10% hexanediol in Figure 5 & 6 does not remove the condensates, but in the earlier figures induced a very significant loss in the number of condensates, not just a loss in intensity. Please explain and justify this discrepancy.

The treatments in Figures 5 and 6 were different from the ones in earlier figures. Figures 5 and 6 are non-treated with salt stress whereas previous Figures 2 - 4 analyzed osmotic stress-induced insulator bodies. The visible large insulator bodies such as the ones showed in figures 2 - 4 are not present under physiological, non-salt stress, conditions. Hence the reliance on intensity changes in Figure 5 and 6. Earlier studies show that insulator proteins form small speckles constitutively, which in our lab are only visible after deconvolution (see Schoborg et al 2013, JCB). The large, more visible, insulator bodies are osmotic stress induced.

Our intuition is that different conditions change the material properties of the insulator protein condensates and hence their variable sensitivity to 1,6-hexanediol (for example, by altering the relative contribution of weak electrostatic versus hydrophobic interactions in the formation of the condensate). Another explanation could also be that at physiological conditions, strong DNA-protein interactions are more predominant than the diffusion seen during salt stress in insulator bodies. This is especially true if one considers the fact that at salt stress, the insulator bodies seem to be non-DNA bound though they are still within the nucleus.

To limit the usage of 1,6-hexanediol and to avoid confusion, in the new version of the manuscript we are only presenting its effect under non-osmotic stress conditions on polytene chromosomes (Figure 5).

In Figure 8, there seems to be a labeling mistake in panel C, it is supposed to be H2Av, without gamma, no? In that case, my question is: what you are seeing for H2Av is not really that it is not lost upon hexanediol treatment, but rather that it was not even there at a high level to begin with? For gamma-H2Av I agree with your description of a loss from the bands, the intensity is quite high in the untreated case. But for H2Av, it seems rather low in both conditions, as if it is only present at some sites at low levels. Also, could this imply that gamma-H2Av is experiencing a combination of DNA-binding with LLPS amplifying this effect, while H2Av is only experiencing DNA-binding?

The reviewer is right. There was a labelling mistake in the previous panel 8C. Multiple quantification of the intensity measurements between H2Av and gamma-H2Av 1,6-hexanediol sensitivity show the same trend. We are however not presenting H2Av and gamma-H2Av 1,6-hexanediol sensitivity in this manuscript as the analyses seem to be an over-stretch of the usage of 1,6-hexanediol.

Results show that, Rad21::Myc overlaps substantially with CP190 foci, implying that the cohesin subunit Rad21 is also involved in the formation of insulator bodies." This conclusion mixes colocalization (measured in the experiment) with a contribution to the condensate formation (stated by the authors). This conclusion cannot be made from colocalization data. The conclusions on the functional interaction between insulator bodies and the cohesion subunit Rad21 are not sufficiently supported by the hexanediol experiments. These experiments are too unspecific to the molecular interaction, the structural basis of this interaction, as well as functional aspects that are alluded to.

We have reworded the Rad21 section to depict colocalization rather than Rad21 contributing to insulator body formation, jus emphasizing the fact that both cohesin and Insulator proteins coalesce both at the polytene insulator bands and at the insulator bodies, which as we have shown have LLPS properties.

It is not described how raw data are or will be made available upon publication. Raw data and, where existing, analysis scripts/codes must be made publicly available. "Upon reasonable request" is insufficient.

Raw data will be made publicly available after acceptance of the manuscript for publication for example in the GitHub repository (bamankwa). As well as in a yet undetermined repository for microscope images.

Statistical reporting and repeat observation reporting is insufficient. Number of analyzed cells (n), number of independent biological (not technical) repeats (N) must be stated, as well as exact P values. For "representative" or "typical" images, it must be stated how many times the

Full Revision

overall experiment was repeated, and in how many cells / animals the observed phenomenon could be confirmed by the experimenter (if not in all, please state fraction, count, or percentage). For each genotype or treatment, three correlated biological replicates were combined. Where necessary, the number of analyzed cells and images taken for each treatment are stated in the figure legends. The exact p-values are also stated in the figures.

All minor concerns of reviewer 1 most of which required rewording have been addressed as shown below.

MINOR CONCERNS

Introduction

The introduction is very well-written, gives a comprehensive but succinct update on the background, and then clearly outlines the question addressed as well as the main findings. I have a number of suggestions how to improve the introduction, but these should be understood as improvements to an already excellent text.

That chromatin 3D organization is "essential" for biological function is not universally accepted. Especially on the level of large-scale organization (compartment A/B domains, territories), it is not really clear whether these serve a general function.

We have reworded the text and removed the usage of the word 'essential' to address the not-so-accepted function of the large-scale genome organization

„thereby stabilizing certain enhancer-promoter interactions", here a word like "favoring" or "supporting" might be better chose than "stabilizing". The reason is that stabilizing, to an extent, implies a quite stable structure/contact, but the E-P contacts are actually rather sporadic, and also the effect of TAD boundaries on these contacts is somewhat subtle. I do not say that the effect of insulation is functionally irrelevant (far from it), but the idea of a contact that is present in the majority of cells in a population in most cases seems to be not in line with observations. A choice of words that better fits this transient nature of contacts might be preferred by the authors.

We have changed "stabilizing" to "favoring".

Where a number of articles are cited in support of functional relevance of TAD boundary insulation, it would be good to also cite Lupianez et al. Cell 2015. This paper was well ahead of its time and scattered most doubts that the TAD structure is irrelevant to cell and organism.

The Lupianez et al. Cell 2015 reference has been included

„Even though strong genomic distribution overlaps have been shown between IBPs and cohesin proteins in *Drosophila*, no data demonstrates a contribution of DNA loop extrusion in its spatial genome organization." This sentence is a bit hard to understand, not exactly sure what it should mean. It seems important, though, so maybe rephrase for clarity?

„It is also worth noting that, even in mammals, not all TADs can be explained by the loop extrusion model." This point was already made for contact domains in Rao et al. 2014, so for fairness it should also be cited here. They put convergent CTCF motifs at about 60% of observed domains (check this number to be safe), the other percent are assigned as likely to be caused by loops between "sticky protein accumulations".

The reference for Rao et al. 2014 has been inserted in the text

"LLPS mediates the formation of a myriad of biological condensates including the nucleolus, stress granules, paraspeckles and p-bodies" if original literature is cited for phase separation examples, I would recommend to go for the observations that established these phenomena. This should certainly include the Brangwynne et al. 2009 Science and 2011 PNAS papers.

„several lines of evidence indicate that phase separation modulates the segregation of the eukaryotic genome into active and inactive compartments." One should also include Falk et al. Nature 2019 here.

The reviewer's recommended references have been inserted

„This is supported by ... the super- enhancer driven association between transcription factors, the mediator complex and RNA polymerase." I do not share this understanding. I will explain. The association between SEs, mediator (and other factors) and Pol II into clusters can lead to contacts between multiple genomic elements, leading to the often-quoted regulatory hubs, or transcription factories. These structures, however, do not seem to contribute to the establishment of large-scale nuclear compartments (active / inactive compartment). These compartments are established at a very coarse scale of organization (100 nm and above), and the regulatory hubs seem to rather be placed within this landscape without influencing its overall appearance much. Yes, long-range contacts will emerge, but these are rather of the character of very precise, locus-specific modifications of 3D organization, not modifiers of large-scale organization. I recommend to disentangle compartmentalization vs. regulatory hub formation in the introduction. A good review paper to get this right is Kempfer & Pombo, Nature Reviews in Genetics, 2020.

This sentence has been changed to “The regulatory hub formation of super-enhancers, transcription factors, the mediator complex and RNA polymerase are also proposed to be LLPS driven” and the appropriate references inserted.

The explanation of the effect of coalescence of IBPs upon salt (osmotic?) stress is a bit confusing. The authors say that long-range interactions are reduced, and in the next sentence they say the condensation introduces long-range interactions. I guess it works in the way that the IBPs coalesce in a chromatin-depleted structure, so they no longer mediate long-range contacts, but admittedly I am left somewhat confused. Also, do all IBPs participate in this? Would it be important to mention which specific ones?

The paragraph sought to decouple insulator protein function from the effect of their assembly into bodies. Whereas literature still supports the long-range function of IBPs our previous work points out that insulator bodies do not serve as contact sites for organizing the Drosophila 3D genome. This is because the fraction of DNA bound insulator proteins is significantly reduced as they migrate to insulator bodies in the nucleoplasm. We have reworded the text to make these points clearer.

„insulator bodies have not been subjected to the hallmarks of LLPS and whether IBPs under physiological conditions assemble through this process is not known." Please write "assessed for hallmark features that support LLPS, so that it remains unknown whether IBPs form insulator bodies via LLPS under physiological conditions." Or something to that effect. My point is, to "subject something to hallmarks" makes logically no sense - even though it is clear what you mean to say by this, please just correct the language here.

We have changed the text to “assessed for hallmark features that support LLPS, so that it remains unknown whether IBPs form insulator bodies via LLPS under physiological conditions” as suggested by the reviewer.

Other minor comments

Figure 1 D phase diagram regions are not color-blind friendly.
Throughout all micrographs, red & green is not color-blind friendly. Please modify to be color-blind friendly.

Images have been made color blind friendly by changing all red channels to magenta.

How was circularity determined on a technical level?

We have spelt out how circularity was determined in the updated manuscript. In brief, “ The circularity index of insulator bodies was calculated by $4\pi A/C^2$ where A is the area of the insulator body mask and C is the perimeter of the insulator body mask. These calculations were

done with FIJI. Circularity value of 1.0 indicates a perfect circle and an approach towards 0.0 as an increasingly elongated polygon”.

Figure 3, I do not understand how a single time point can represent a time span of 10 seconds, e.g. 0-10 s is indicated in the example image, and also in panel C.

In figure 3, images were taken in 10seconds intervals for 3 minutes.

The style jumps for how figures are referenced, for example (Figure 1A) and (fig. 1A) both occur.

Uniform styles are adopted in the updated manuscript

REVIEWER 2

Additional LLPS properties were not analyzed, nor discussed or evaluated in comparison to the prediction of IDRs, electrostatic forces, hydrophobic interactions, though available algorithms are available for this purpose. Thus, properties such as prediction of Prion-like domains, low-complexity regions, coiled-coiled structures, should be considered for the first part of the manuscript. Alternatively, a detailed discussion should highlight a possible correlation with the parameters analyzed in Figure 1.

Judging from the fact that LLPS induction mechanisms are quite heterogenous and each protein would depend on specific stimuli to undergo condensation w,e agree with the reviewer suggestion to introduce additional parameters in our insulator protein analysis. The new version of the manuscript introduces additional LLPS prediction algorithms which we consider more relevant to our proteins of interest. These include Pscore and LARKS (low-complexity aromatic-rich kinked segments). Based on these analyses, we resorted to further investigating the charged residues as they show unique patterns in the insulator proteins significantly extending this results section and the discussion.

-The role of RAD21 is weakly verified in comparison to the insulator and architectural proteins. More evidence for its role in this molecular model is required. For instance, further, its distribution in the experiment shown in Figure 4A, since its colocalization with Cp190 in polytene chromosomes in Supplementary Figure 6 could also suggest redundant functions.

It is true other non-tested proteins including RAD21 could be redundant scaffolds with CP190 and Mod(mdg4)67.2. However, because cohesin and insulator proteins work together in the current mammalian model of genome organization, we only intended to ask whether any of the

cohesin subunits was also colocalizing with insulator proteins in insulator bodies. Further analysis of the cohesin role in the condensate is beyond the scope of this manuscript. For example, we are interested in a model where both cohesin and insulator protein function depend on the LLPS properties of their proteins, but this is an early stage of this investigation. As highlighted by reviewer number 1, the wording was confusing in this section of the manuscript. We have hence rephrased this section to emphasize our finding that a cohesin subunit is a component of insulator bodies.

The authors repeatedly make inferences in the context of 3D genome organization. However, this manuscript lacks experimental validation using chromosome conformation technologies such as HiC. Therefore, they should avoid overinterpretation of results or to discuss them in the context of TADs resolution in *Drosophila*, as previously reported (PMID: 28760140).

*We agree with the reviewer and we have reworded the manuscript to avoid overinterpretation of results. However, a few of the inferences are made to imply that, giving our results, the LLPS properties of insulator proteins may be biologically significant and they may play a role in the organization of the *Drosophila* genome. Future research in our lab will test this potential role using tools such as Hi-C and other genomic approaches. But we agree with the reviewer that in the absence of this experiments the implication of our results should be taken cautiously.*

-From the photobleaching assay in polytene chromosomes, might the band recovery of insulator proteins be associated with their specific topology in subnuclear structures (P bodies, speckles...). Did authors analyze their colocalization with specific markers of subnuclear structures?

*In our previous work and through an examination of the literature we discarded that insulator bodies or insulator protein distribution overlaps or associates with other known condensates including speckles (see Schoborg et al 2013, JCB). Additionally, we have used the Gal4 driven expression of *Su(Hw)* in different specific tissues and have shown that the *Su(Hw)::GFP* can rescue null *su(Hw)* induced phenotypes like sterility and *cut*⁶. This suggest that ectopic expression of *Su(Hw)::GFP* does not significantly interfere with other structures in the cell, at least not to the point of been deleterious. In polytene chromosomes we have live imaging showing how *Su(Hw)::GFP* bands are unambiguously visible and respond to osmotic stress conditions by forming insulator bodies just like in diploid cells (see Schoborg et al 2013, JCB) . It would be impossible to completely discard that unknown condensates are associating with *Su(Hw)::GFP* in our experiments, but given how *Su(Hw)::GFP* recapitulates function and distribution of the endogenous *Su(Hw)* proteins, we believe that an effect of unknown condensates in our FRAP experiments is very unlikely.*

-Authors mentioned along with the explanation for Figure 2 that 1,6-hexanediol responsiveness changes in recovery experiment as shown by Schoborg et al., 2013. Did they perform any

recovery experiment that verifies such preliminary observations? If not, elaborate more on the implications of this statement on their own results.

We meant recovery from salt stress treatment as reported in the Schoborg et al., 2013 paper and not recovery from the hexanediol experiment. We sought to make a distinction between the behavior of insulator proteins in recovery from salt stress and in the hexanediol treatment (not hexanediol recovery). During salt stress recovery, insulator bodies dissolve and insulator proteins repopulate chromatin. Similarly, treatment with hexanediol also dissolves insulator bodies, but interestingly insulator proteins do not repopulate chromatin. Hexanediol recovery is not equivalent to salt recovery because recovery from hexanediol would imply reconstitution of insulator bodies. The implication is that treatment with hexanediol does not favor the return of proteins to chromatin and our speculation is that proteins do not return because LLPS is also necessary to maintain association of Insulator proteins to chromatin..

In Figure 2, a Ctrl/PBS condition is necessary to show the non-stimulated distribution of Cp190 and Su(Hw). Why Mod(mdg4)67.2 was not included in this panel?

Reviewer is right about the need to include a non- stimulated distribution of Cp190 and Su(Hw) controls in figure 2. However, the non-stimulated distribution of CP190 and the other insulator proteins compared to their salt stress responses are well documented in our previous paper (Schoborg et al., 2013, JCB) and elsewhere,. We do have data on Mod(mdg4)67.2 as a marker as well. We thought including Mod(mdg4)67.2 in this analysis would be redundant given that all insulator proteins are targeted to the same condensates.

Based on their cells with mutant backgrounds for Cp190 or Mod(mdg4)67.2, authors conclude that they act as redundant scaffold proteins for the formation of insulator bodies. What is known about their direct interaction? Is there any published evidence with specific, differential, interactors between them? This information should be added to the Discussion.

We agree with the reviewer this information is important in the context of these proteins' ability to condensate. Missing this information in our paper was an oversight on our part. We have included wording and references on the well established data about interactions between CP190 and Mod(mdg4)67.2 as well among other insulator proteins in the discussion.

Figure legends and panels have remarkable inconsistencies, which I will indicate separately:

*Figure 1. Panel B corresponds to C and vice versa. Current panel B does not display NCPR on the X-axis. Current panel C should include the N number for each group, and its relation to Supplementary Figure 1. Panel D uses a color code instead of Region A-E; therefore, the figure legend generates confusion.

All inconsistencies in legends and panels have been rectified in the updated version

*Figure 2. Cells used are not specified. Scale is missing but displayed on the IF images.

Full Revision

Drosophila imaginal wing disc cells were used and is now specified. Scale bar is also displayed.

*Figure 3. Quantification of insulator body average size could be as relevant, or more than insulator body number in Panel C. For instance, that reference regarding body size is included on the legend for Figure 4, panel A. Panel B was analyzed at time point 60s, is there any significant difference when compared to 30s? What is referred to as "stressed condition", the label on the figure? Were these experiments made by triplicate? Or is one essay enough to say that insulator bodies undergo fusions? Standard deviation on panels 3B and 3C? Scale is missing.

We thank the reviewer for bringing this to our attention. Insulator body size analysis has been now included in Figure 3 Panel C. Error bars are included in both the insulator body number and size analysis. The length of time for analyzing the fusion has also been extended to make clearer the differences between the time points. In all cases, stress refers to 250mM NaCl. Fusion between insulator bodies has been observed multiple times and in independent experiments, and it has been documented previously in our laboratory (Schoborg et al 2013).

*Figure 4. In Panel A, Mod(mdg4) is red in the Figure and not green. The text mentioned that 1,6-hexanediol was used for the experiment presented on Panel C, however, it is not clear whether all conditions represent 1,6-hexanediol treatment or not, proper labelling is missing.

Panel C of figure 4 looks at the contribution of the individual insulator proteins to the sensitivity of 1,6-hexanediol and to an extension, the material property of insulator body. It turns out all the tested constituents contribute equally to 1,6-hexanediol sensitivity without any significant changes in the number of insulator bodies irrespective of the background tested. We are however not including this panel as it adds little information to this manuscript and appears hard to interpret.

*Figure 5. In panel A, the scale is missing but displayed on the IF images. Without the quantification presented on Panels B and D, Panels A and C are not showing evident differences and the resolution is very poor.

We realize the information in Panels C and D are better answered in figure 6. We are therefore removing panels C and D in figure 5.

*Figure 6, panels D, E, and F. How many replicates are included in these results to be significant? Did Mod(mdg4) show the same pattern upon 10% 1,6-hexanediol treatment?

Triplicates were done for these experiments. As mentioned by reviewer 1, 10% hexanediol may be detrimental to the cells and so we removed 10% hexanediol data from the manuscript. We are only presenting 5% results now. And yes, Mod(mdg4) is also sensitive to hexanediol on

Full Revision

polytene chromosomes, although data is not shown. (Mod(mdg4) is present at every binding site of Su(Hw)).

*Figure 7. The title should be more precise, their conclusion is inaccurate based on the applied statistical analysis. Panel B should be better described, "more striking" is not describing properly their observations. The basal levels of Cp190 compared to dCTCF highly differed yet in the Media/PBS condition, and the standard deviation inside this group is very different between them. Therefore, this conclusion should be more critical.

Since figure 7 had 10% hexanediol treatment, it has been discarded entirely from this version of the manuscript.

June 27, 2022

RE: Life Science Alliance Manuscript #LSA-2022-01536

Dr Mariano Labrador
The University of Tennessee
Biochemistry Cell and Molecular Biology
M407 Walters Life Sciences
1414 Cumberland Avenue
Knoxville, Tennessee 37996

Dear Dr. Labrador,

Thank you for submitting your revised manuscript entitled "Drosophila insulator proteins exhibit in-vivo liquid-liquid phase separation properties". We would be happy to publish your paper in Life Science Alliance pending final revisions necessary to meet our formatting guidelines.

- please upload your manuscript file as an editable doc file
- please upload your main and supplementary figures as single files
- please add a running title, category, alternate abstract/ summary blurb and Twitter handle of your host institute/organization as well as your own or/and one of the authors in our system
- please use the [10 author names, et al.] format in your references (i.e. limit the author names to the first 10)
- please double-check your figure callouts; you have a callout for Figure S6, but this is not in the legend or uploaded
- please add a panel D to your Figure S5 legend

A. FINAL FILES:

B. MANUSCRIPT ORGANIZATION AND FORMATTING:

Sincerely,

Reviewer #1 (Comments to the Authors (Required)):

Authors have either addressed or critically discussed all reviewer's comments in a satisfactory manner. The revised version of the manuscript has significantly improved and represents a significant contribution in the field. Therefore I suggest its approval for publication.

Reviewer #2 (Comments to the Authors (Required)):

I very much welcome the authors' effort to address my comments. The manuscript now looks fine to me, and I can give full support.

A few additional comments. They require no response, the authors can take them into consideration if they want.

Sticking with the 5% HE treatment, for two minutes, is a safer choice, even if the concentration is still quite high. Nevertheless, this results is to be seen in the context of numerous other results presented, which also support the claim of a liquid-phase mechanism being at work, so this comment has been addressed.

The BEAF-32 control is a great idea and works very well as a control for analysis sensitivity.

Fig. 3: Seeing the fusion of the droplets is definitely helping to support the LLPS claim, the additional evidence is excellent. In the current display, the intensities are strongly saturated, maybe a different color map and adjustment might help. The observations are visible, though, so this is more of an aesthetic concern.

On the point of Ostwald ripening, fusion and Ostwald ripening are not mutually exclusive. In fact, it is hard to exclude Ostwald ripening, it should be going on in any liquid phase system. I do agree the fusion are very obvious, and also fusions are proceeding much faster than Ostwald ripening, especially in early phase domain ripening. I would suggest a different wording, the current description does not make sense in the light of phase separation theory. Maybe something like this? "Obviously visible in the data are fusion events via coalescence. Ostwald ripening - the dissolution of small liquid bodies in favour of the growth of larger liquid bodies - is not apparent as a prominent mode of droplet coarsening (84)."

For the deposition of data, I would recommend the Zenodo service. Data of any format can be deposited, and shared with a citable DOI etc. Also, image analysis code and raw data can be shared together on this platform. Code sharing vis GitHub, as suggested by the authors, is of course also a good option.

July 6, 2022

RE: Life Science Alliance Manuscript #LSA-2022-01536R

Mariano Labrador
University of Tennessee at Knoxville
Biochemistry Cell and Molecular Biology
309 Ken and Blaire Mossman Bldg.
1311 Cumberland Avenue
Knoxville, Tennessee 37996

Dear Dr. Labrador,

Thank you for submitting your Research Article entitled "Drosophila insulator proteins exhibit in vivo liquid-liquid phase separation properties". It is a pleasure to let you know that your manuscript is now accepted for publication in Life Science Alliance. Congratulations on this interesting work.

DISTRIBUTION OF MATERIALS:

Again, congratulations on a very nice paper. I hope you found the review process to be constructive and are pleased with how the manuscript was handled editorially. We look forward to future exciting submissions from your lab.

Sincerely,
